# Model-free photon analysis of diffusion-based single-molecule FRET experiments

Ivan Terterov [1] ✉, Daniel Nettels [2], Tanya Lastiza-Male[1,5], Kim Bartels[3,4,6], Christian Löw [3,4], Renee Vancraenenbroeck[1,7], Itay Carmel [1], Gabriel Rosenblum[1] & Hagen Hofmann [1] ✉

Photon-by-photon analysis tools for diffusion-based single-molecule Förster resonance energy transfer (smFRET) experiments often describe protein dynamics with Markov models. However, FRET efficiencies are only projections of the conformational space such that the measured dynamics can appear non-Markovian. Model-free methods to quantify FRET efficiency fluctuations would be desirable in this case. Here, we present such an approach. We determine FRET efficiency correlation functions free of artifacts from the finite length of photon trajectories or the diffusion of molecules through the confocal volume. We show that these functions capture the dynamics of proteins from nano- to milliseconds both in simulation and experiment, which provides a rigorous validation of current model-based analysis approaches.

Probing the dynamics of biomolecules has become a major task of single-molecule Förster resonance energy transfer (smFRET) experiments. Structures alone are insufficient to understand protein function. Instead, timescales and amplitudes of structural changes in enzymes[1–5], transporters[6–8], molecular machines[9,10], and disordered proteins[11–14] are required to understand their biological role. At the level of individual molecules, motions are stochastic and driven by thermal noise. Powerful tools to retrieve dynamics from stochastic trajectories are correlation functions. However, information in smFRET experiments, particularly with freely diffusing molecules, is scarce. Roughly 100–200 donor and acceptor photons are detected in a burst, i.e. during the short millisecond transit of a molecule through the confocal volume of a microscope. Although correlation functions of these short photon traces contain information on structural changes, it is obscured by the finite length of the trajectory and by the diffusion process through the confocal volume itself. Therefore, innovative analysis tools to retrieve this information were developed with the goal of identifying the number and type of structural states of a protein together with the timescales at which they are sampled. Established methods include dynamic photon distribution analysis[15], maximum likelihood methods[16–18] and Hidden–Markov model (HMM) fitting in the form of H²MM[1,19] and mp-H²MM[20]. Despite differences in the details, these methods optimize the parameters of a model given the measured photon trajectory. Models are typically first-order chemical kinetic schemes of photon-emitting conformational states together with kinetic rate constants that describe switching between the states. To test the goodness of a fit, the photon traces are recolored using the model fit and then compared with the experimental data, and/or the Viterbi algorithm is used to check whether the lifetimes of the states in the model are exponentially distributed[10]. In the following, we term these tools model-based approaches because a kinetic model is a prerequisite for the analysis.

A drawback of these model-based approaches is that the model choice is not always obvious from the experimental FRET efficiency histograms. Several models need to be checked and a compromise between over-fitting and fit quality must be found. Often neglected, the FRET efficiency is a projection of the high-dimensional coordinate space of protein structures onto a single coordinate. Not all motions of a protein will necessarily cause a change in FRET efficiency, which can lead to apparent non-Markov behavior[21–23] that might be missed by imposing

[1]Department of Chemical and Structural Biology, Weizmann Institute of Science, Rehovot, Israel. [2]Department of Biochemistry, University of Zurich, Zurich, Switzerland. [3]Centre for Structural Systems Biology (CSSB) DESY, Hamburg, Germany. [4]European Molecular Biology Laboratory Hamburg, Hamburg, Germany. [5]Present address: Department of Physics, University of Illinois, Urbana-Champaign, Urbana, IL, USA. [6]Present address: University Medical Center Hamburg Eppendorf, Hamburg, Germany. [7]Present address: Department of Structural & Molecular Biology, University College London, London, UK. ✉e-mail: ivan.terterov@weizmann.ac.il; hagen.hofmann@weizmann.ac.il

Markov models in the first place. An example is the enzyme QSOX (quiescin sulfhydryl oxidase) that samples two macroscopic structural states with power-law kinetics[2]. Model-free methods to extract dynamic information, e.g., in the form of correlation functions, are therefore desirable. These functions provide timescales even via simple visual inspections without necessarily imposing a model. In addition, they ease model identification for model-based analysis approaches and provide an additional test of the adequacy of a model. Current model-free methods to probe conformational dynamics in single-molecule bursts include lifetime-filtered FCS (fFCS)[24,25], two-dimensional fluorescence lifetime correlation spectroscopy (2D-FLCS)[26–28], time-resolved burst variance analysis (trBVA)[29], and recurrence analysis of single particles (RASP)[13,30]. In these methods, the experimental photon traces are pre-processed to extract timescales of FRET efficiency fluctuations. Unfortunately, most of them require very long measurements to reach sufficient signal-to-noise (fFCS, 2D-FLCS, RASP) or they provide dynamic information in a rather indirect manner (trBVA).

Here, we present a simple but effective alternative. We show how to compute the autocorrelation function of FRET efficiencies[31] (hereafter referred to as FRET correlation function) free of artifacts due to the finite length of the photon trace and the diffusion through the confocal volume. Using realistic simulations of smFRET experiments of diffusing molecules, we demonstrate that the timescale of FRET efficiency fluctuations can be correctly identified from microseconds up to milliseconds. Using experiments on a DNA-based Holliday junction and a membrane protein, we obtain dynamic information from as few as a few thousand molecules. We also show how this tool can be extended to probe the sub-microsecond dynamics of IDPs in nsFCS experiments[32–35].

## Results

### Calculating FRET correlation functions

In smFRET experiments of freely diffusing molecules, the transit of a molecule through the confocal volume results in a burst of donor and acceptor photons (Fig. 1a). Unfortunately, a burst is only a few milliseconds long at best. If the burst duration $T$ is close to the timescale of conformational dynamics ($\tau_D$), problems arise when attempting to extract $\tau_D$ from correlation functions. The problem of computing correlation functions from continuous finite time series has been analyzed by Zwanzig[36], who showed that the relative statistical error scales with $\sqrt{2\tau_D/MT}$ where $M$ is the number of trajectories. An accuracy of 10% in the correlation function of a single molecule would require a burst that is 200 times longer than $\tau_D$. With 200 trajectories on the other hand, even slow dynamics in the order of the burst duration ($\tau_D = T$) might be accessible. Diffusion-based smFRET experiments with thousands of molecules should therefore be sufficient to determine correlation functions even for dynamics comparable to the burst duration. Yet, bursts are not continuous signals but rather streams of photons and their arrival times. We therefore distinguish three FRET efficiencies: the apparent FRET efficiency $E$, defined by the raw photon counts recorded by the donor and acceptor detectors, the FRET efficiency $\tilde{E}$, computed from the photon counts corrected for background, relative dye brightness and instrumental imperfections, and the true FRET efficiency $\varepsilon$ that depends on the donor-acceptor distance $r(t)$ and the dye-specific Förster-distance $R_0$ (Supplementary Note 1). To define experimental correlation functions, we assign an apparent FRET efficiency $E = 1$ to each acceptor photon and $E = 0$ to the donor photons. With $\tau$ being the time between two arbitrary photons in a burst, we define the FRET correlation function as

$$g_E(\tau) = N(\tau)^{-1} \sum_{\substack{photon\ pairs}} \left(E_t - \langle E \rangle\right)\left(E_{t+\tau} - \langle E \rangle\right) \tag{1}$$

Here, $(E_t, E_{t+\tau})$ indicates photon pairs separated by lag times from $\tau$ to $\tau + \Delta\tau$, $N(\tau)$ is the number of all such pairs, and $\langle E \rangle$ is the average

apparent FRET efficiency computed from the photons of all bursts. Importantly, at ideal instrumental conditions, the FRET correlation function $g_E(\tau)$ defined in Eq. 1 is identical with the true correlation function $g_\varepsilon(\tau)$ (Supplementary Note 2). Equation 1 can be reformulated. If $N_{XY}(\tau)$ is the number of photon pairs of type $X$ and $Y$ ($A$ for acceptor and $D$ for donor) separated by the time $\tau$ (Fig. 1b), we can write Eq. 1 as

$$g_E(\tau) = (1 - \langle E \rangle)^2 \frac{N_{AA}(\tau)}{N(\tau)} + \langle E \rangle^2 \frac{N_{DD}(\tau)}{N(\tau)} - \langle E \rangle(1 - \langle E \rangle) \left[\frac{N_{AD}(\tau)}{N(\tau)} + \frac{N_{DA}(\tau)}{N(\tau)}\right] \tag{2}$$

The fractions $N_{XY}(\tau)/N(\tau)$ are related to the ratios of the conventional intensity correlation functions (see "Methods")

$$\frac{N_{XY}(\tau)}{N(\tau)} \approx \frac{\langle n_X(t)n_Y(t+\tau) \rangle}{\langle n(t)n(t+\tau) \rangle}, \tag{3}$$

where $n_X$, $n_Y$, and $n$ are the raw emission rates for photons of type $X$, $Y$, and all photons, respectively. Equation 3 is approximate because the pair numbers $N_{XY}(\tau)$ are computed from bursts with a finite duration but intensity correlation functions such as $\langle n_X(t)n_Y(t+\tau) \rangle$ are typically computed for the whole measurement, including the signal between bursts. While $N_{XY}(\tau)$ and $N(\tau)$ include intensity fluctuations due to the path of the molecule through the inhomogeneously illuminated confocal volume (Fig. 1c, d), these fluctuations cancel to large extent in Eqs. 2–3 (Supplementary Note 3). Similarly, bias due to the finite burst length[37] is marginal (Supplementary Note 4). Hence, Eq. 2 provides a good estimate of the FRET correlation function (Fig. 1d). Notably, a residual impact of diffusion on $g_E(\tau)$ will remain due to dye saturation and different detection volumes for donor and acceptor photons (Supplementary Note 3, Supplementary Fig. 1). However, while these effects indeed impact the $N_{XY}(\tau)/N(\tau)$ ratios, we show that they are negligible in $g_E(\tau)$ (see "Beyond the Poisson limit" and Supplementary Note 3).

### FRET correlation functions in the Poisson limit

To demonstrate the idea of our approach, we performed Brownian dynamics simulations of proteins at constant concentration that diffuse through a confocal volume and switch between two conformational states with the true FRET efficiencies $\varepsilon_1 = 0.1$ and $\varepsilon_2 = 0.9$ and the rate constants $k = k_{12} = k_{21}$ (hereafter referred to as rates) (Fig. 2a). We added background photons, differences in the brightness of donor and acceptor, and the possibility to directly excite the acceptor by the donor excitation laser (see "Methods"). The simulations assumed Poisson photon emission statistics that is correct at sufficiently low excitation rates and at timescales slower than the fluorescence lifetimes of the dyes. At low switching rates, the FRET histograms show two defined peaks (Fig. 2b). With increasing rates, intermediate FRET values become prominent because more molecules change their conformation during the transit through the confocal volume. At very fast exchange, the two peaks finally merge into a single peak at intermediate FRET efficiency, thus giving the impression of a single conformational state[38]. We then computed FRET correlation functions using Eq. 2. These functions exhibit monotonic decays (Fig. 2c). The scatter in these decays increases with increasing lag-time due to the lower number of photon pairs that contribute to the correlation function at long times (Fig. 1c,d). Exponential fits describe the decays well, as expected for a 2-state model (Fig. 2c). The apparent relaxation rate ($\lambda$) is the nonzero eigenvalue of the rate matrix, which is given by $\lambda = k_{12} + k_{21}$, and the apparent relaxation time is expected to be $\tau_D = 1/\lambda$. A comparison of the true relaxation times with those determined from fits of the FRET correlation function shows excellent agreement (Fig. 2d). Notably, the correlation functions are exact, i.e., the value near zero lag time

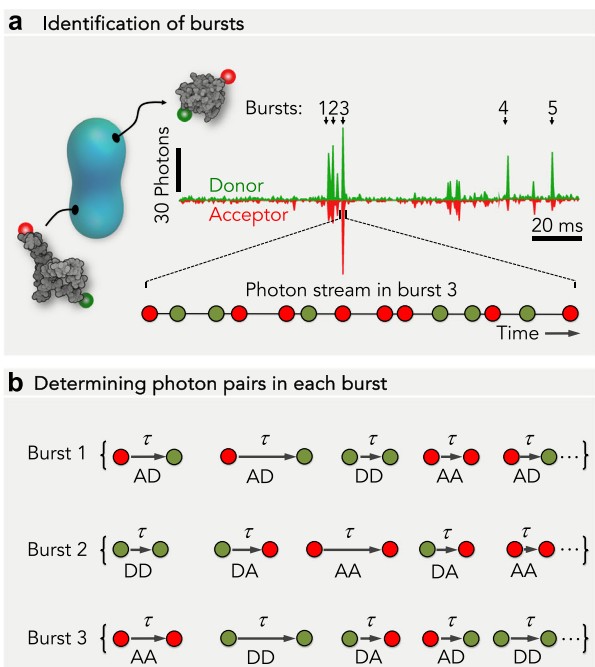

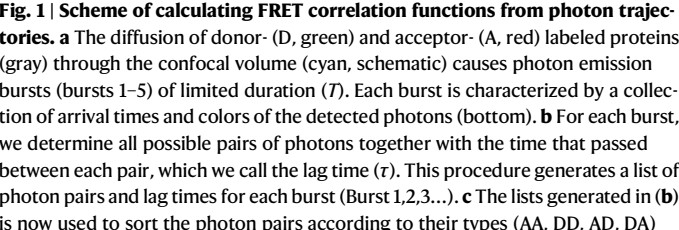

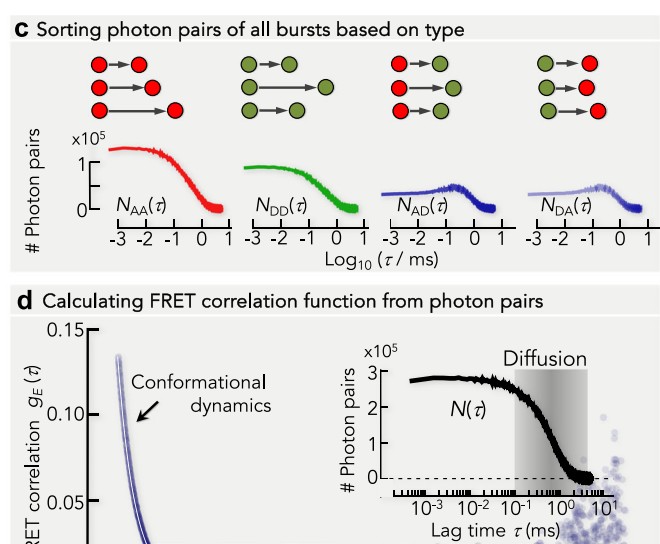

**Fig. 1 | Scheme of calculating FRET correlation functions from photon trajectories. a** The diffusion of donor- (D, green) and acceptor- (A, red) labeled proteins (gray) through the confocal volume (cyan, schematic) causes photon emission bursts (bursts 1–5) of limited duration ($T$). Each burst is characterized by a collection of arrival times and colors of the detected photons (bottom). **b** For each burst, we determine all possible pairs of photons together with the time that passed between each pair, which we call the lag time ($\tau$). This procedure generates a list of photon pairs and lag times for each burst (Burst 1,2,3...). **c** The lists generated in (**b**) is now used to sort the photon pairs according to their types (AA, DD, AD, DA) irrespective from which burst a photon pair originated (top). A histogram of the photon pairs of each photon pair type ($N_{AA}(\tau)$, $N_{DD}(\tau)$, $N_{AD}(\tau)$, $N_{DA}(\tau)$) is constructed (bottom, schematic). The sum of these four histograms gives the histogram of all photon pairs, irrespective of color: $N(\tau) = N_{AA}(\tau) + N_{DD}(\tau) + N_{AD}(\tau) + N_{DA}(\tau)$ **d** Dividing the photon pair histograms by the histogram of all photon pairs $N(\tau)$ (inset), provides the correlation ratios ($N_{AA}(\tau)/N(\tau)$, $N_{DD}(\tau)/N(\tau)$, $N_{AD}(\tau)/N(\tau)$, $N_{DA}(\tau)/N(\tau)$). Using Eq. 2, the FRET correlation function is computed (blue circles, schematic). The timescale of diffusion is indicated as gray shaded area.

quantifies the variance of the FRET fluctuations. In our 2-state system, an approximate expression for the amplitude is $g_E(0) = p_1 p_2 (E_2 - E_1)^2$ where $p_1 = p_2 = 1/2$ are the relative populations of the two states and $E_1$, $E_2$ are the apparent FRET efficiencies of the two states. All functions decay to zero at long lag-times. However, static heterogeneity, e.g., due to a mixture of states that do not interconvert or that interconvert at timescales much slower than the diffusion time through the confocal volume $t_D \sim 1$ ms, would manifest as an offset in the correlation functions (Fig. 2e). This is a particularly advantageous property. In fact, static heterogeneity is difficult to spot otherwise, except when comparing of FRET efficiencies with the fluorescence lifetimes of the dyes[39–41], and it is rarely included in model-based analysis approaches.

**The effect of protein concentration on the correlation functions**
Like in actual smFRET experiments, we simulated the data in Fig. 2a at the low concentration of 50 pM to ensure that the chance of simultaneously observing two or more molecules in the confocal volume is negligible. However, in some cases it might be necessary to perform smFRET experiments at higher concentrations, e.g., in inter-molecular FRET experiments if the affinity of the binding partners has to be matched with the concentrations of labeled molecules, or if many photons are required such as in sub-population resolved nsFCS experiments[32,42,43]. Assuming a confocal volume of 1 fl, Poisson statistics shows that 1.5% of the bursts include more than one molecule at a concentration of 50 pM. This fraction increases to 6% at 200 pM. These "mixed" bursts unavoidably affect FRET correlation functions because two or more molecules in the confocal volume will average the FRET fluctuations[44]. We tested this effect by performing simulations at different protein concentrations. To exclusively study the concentration

effect, we used a system in which two states with equal populations do not interconvert (static heterogeneity). At infinite dilution, the correlation function is expected to be a flat line at $g_E(\tau) = (E_2 - E_1)^2/4$ that quantifies the static heterogeneity of the mixture. Yet, with increasing protein concentration, $g_E(\tau)$ decays due to the de-correlation caused by multiple protein molecules in the confocal volume (Fig. 2e). The timescale of this decay is determined by the concentration of the molecules and is rather long (>10 ms for concentrations <100 pM) (Fig. 2d, f). To avoid misinterpretations, we therefore suggest experiments at the lowest possible protein concentration for quantifying static heterogeneity or very slow dynamics. Notably, this concentration effect is inherent to the experiment and will affect every analysis method.

**Non-Markov processes and model validation**
FRET efficiency fluctuations are 1D-projections of motions in a high-dimensional space spanned by the atomic coordinates of the protein. It can never be excluded that proteins explore states that are indistinguishable on the FRET efficiency coordinate. A recent example is Hsp90[45]. The dynamics of FRET efficiency fluctuations in this case will be non-Markovian with non-exponential dwell time distributions of the distinguishable states. Yet, analyzing dwell times first requires the assignment of states in the trajectory. While this is straightforward when photon fluxes are high, it is a daunting task in smFRET experiments of diffusing molecules. A few hundred photons per millisecond are insufficient to identify states without a model, which is the very idea behind model-based analysis tools. Yet, imposing Markov models on experimental data with potential non-Markov dynamics could be problematic. We demonstrate this aspect in a simulation. We first simulated a 2-state model with FRET efficiencies $\varepsilon_1 = 0.2$ and $\varepsilon_2 = 0.8$

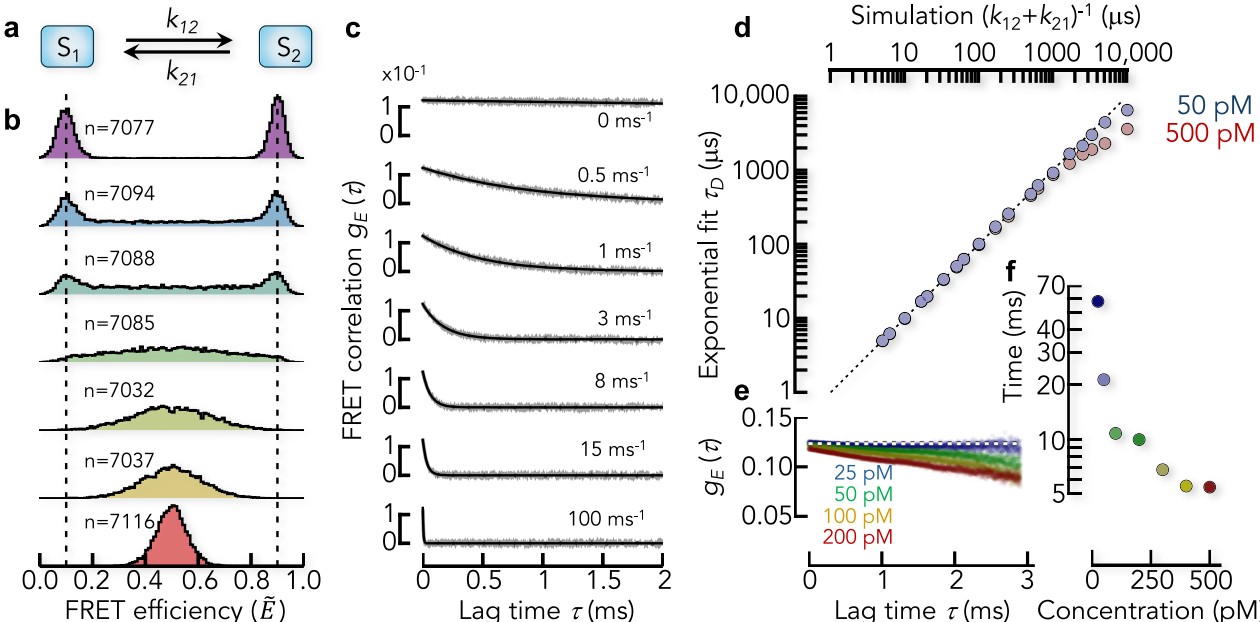

**Fig. 2 | Determining dynamics from FRET correlation functions. a** Kinetic scheme of the 2-state model. **b** FRET efficiency histograms (corrected) from Brownian dynamics simulations including the photon emission process of donor and acceptor. The FRET efficiencies of the two states in (**a**) are indicated by dashed lines. The total number of bursts is indicated for each histogram. The kinetic rates $k = k_{12} = k_{21}$ (indicated in **c**) increase from top to bottom. The average burst duration was $t_D = 1.2$ ms. **c** FRET correlation functions computed from the data in (**b**). The black line is a fit with an exponential decay and the apparent rate $\lambda$. The kinetic rate used in the simulation is indicted. **d** Comparison between simulation and the

analysis using FRET correlation functions. The total relaxation time from the exponential fits of the FRET correlation functions ($\tau_D$) is compared with the expected value $\lambda^{-1} = (k_{12} + k_{21})^{-1}$. The results of simulations with two protein concentrations (indicated) are shown. **e** FRET correlation functions for a static 2-state model ($k = 0$) at different protein concentrations. The theoretical value of the FRET correlation function at infinite dilution is indicated by the dashed line. **f** Decay times of the FRET correlation functions in (**e**) after fits with an exponential decay.

and the rates $k = 5\,\mathrm{ms}^{-1}$, which results in a broad FRET efficiency distribution centered between both states (Fig. 3a). A model-based analysis (see "Methods") using the correct 2-state model provides an excellent fit as judged by a recoloring with the 2-state model (see "Methods") and it reliably retrieves the simulated rates together with the FRET efficiencies of states 1 and 2. When using the recolored photon traces to compute the model-derived FRET correlation function, the original and model-derived functions are in excellent agreement, as expected (Fig. 3b). Now, we render the system non-Markov by letting each state convert to a "mirror-image" state (1′ and 2′) with the same FRET efficiency as the original state, leading to a 4-state model that appears 2-state on the FRET efficiency coordinate (Fig. 3c). For simplicity, we assume all rates to be identical and we use the same value as in the 2-state simulation ($k = 5\,\mathrm{ms}^{-1}$. The simulation now shows a broad distribution of FRET efficiencies with peaks at $\widetilde{E}_1 = 0.2$ and $\widetilde{E}_2 = 0.8$ and a distribution of bursts in between (Fig. 3c).

The change in histogram compared to the 2-state simulation (Fig. 3a) is not surprising as the system now spends more time at either FRET efficiency, which causes a slower apparent rate of switching between high and low FRET states. Despite the increased complexity, a model-based analysis with a 2-state Markov-model provides an excellent fit of the data and there is no indication that the model might be inappropriate based on recoloring (Fig. 3c). The BIC (Bayes Information Criterion)[46] is often used to determine the quality of a fit in Hidden-Markov modeling. It is given by $BIC = R\ln T - 2\mathcal{L}(\mathrm{m})$, where $R$ is the number of free parameters in the fit, $T$ is the length of the trajectory, and $\mathcal{L}$ is the maximized value of the log-likelihood function of the model m (see "Methods"). In our case (Markov vs. non-Markov), the BIC of fits with the 2-state-model to both data sets can only differ by $\mathcal{L}$. We found $\mathcal{L} = 16.8$ for the fits of both data sets, thus providing no chance to distinguish the fit quality. Falsifying the model choice either requires a dwell-time analysis or additional model-independent

information. Indeed, when we compared the FRET correlation function of the data with that obtained from the 2-state Markov-model fit, we found a substantial discrepancy (Fig. 3d). The correlation function of the data is non-exponential whereas the two-state fit results in a single-exponential decay. This comparison clearly invalidates the simple 2-state model in the case of non-Markov effects, thus calling for a more complex model.

### Identifying photobleaching with correlation ratios[47]
The FRET correlation estimate can be decomposed into four correlation ratios (Eq. 2) that fulfill the condition

$$1 = \frac{N_{AA}(\tau)}{N(\tau)} + \frac{N_{DD}(\tau)}{N(\tau)} + \frac{N_{AD}(\tau)}{N(\tau)} + \frac{N_{DA}(\tau)}{N(\tau)} \qquad (4)$$

We define the normalized correlation ratios as

$$f_{AA}(\tau) = \frac{1}{\langle E \rangle^2} \frac{N_{AA}(\tau)}{N(\tau)}, f_{DD}(\tau) = \frac{1}{(1 - \langle E \rangle)^2} \frac{N_{DD}(\tau)}{N(\tau)}, \text{and}$$

$$f_{AD/DA}(\tau) = \frac{1}{\langle E \rangle (1 - \langle E \rangle)} \frac{N_{AD/DA}(\tau)}{N(\tau)}.$$

The correlation ratios contain similar information as $g_E(\tau)$ on the dynamics, but they are more prone to artifacts (see next section). Yet, they are helpful in identifying acceptor photobleaching due to chemical reactions of the dye with reactive oxygen species in the solution. To demonstrate this aspect, we performed simulations of a two-state system without and with acceptor photobleaching (Fig. 4a) and then computed the correlation ratios (Fig. 4b) and FRET correlation functions (Fig. 4c). In the absence of photobleaching, the correlation ratios are symmetric with respect to a lag time of $\tau = 0$ ms (Fig. 4b left). Yet, when the acceptor of a molecule photobleaches during the transit

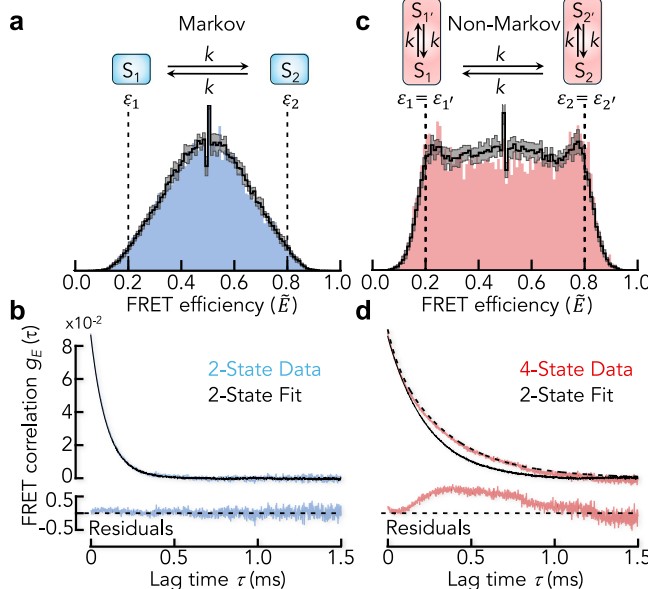

**Fig. 3 | Model validation using FRET correlation functions. a** FRET efficiency histogram (blue) of a simulation using a 2-state model (schematic) with inter-conversion rates $k = 5\,\text{ms}^{-1}$. Black line is the average of 10 recolored data sets based on a fit of the data with a 2-state Hidden-Markov model. The gray shaded band indicates the mean ± SD of the 10 realizations. The fitted rates are $5.1\,\text{ms}^{-1}$ for the forward and backward reaction. **b** FRET correlation function (top) of the original data (blue) and the recolored data (black) and the corresponding residuals of the fit (bottom). **c** FRET efficiency histogram (red) of a simulation using a 4-state model (schematic) with the rates $k = 5\,\text{ms}^{-1}$. Black line and gray shaded area as in (**a**). The fitted rates with the 2-state model are $2.1\,\text{ms}^{-1}$ for the forward and backward reaction. **d** FRET correlation function (top) of the original data (red) and the recolored data (black) and the corresponding residuals of the fit (bottom). The black dashed line is the analytically calculated FRET correlation function of the non-Markov model in (**c**).

through the confocal volume, the density of donor photons will be higher towards the end of a burst than at its beginning. The cross-correlation ratios $f_{AD}(\tau)$ and $f_{DA}(\tau)$ will therefore not be identical (Fig. 4b middle). Acceptor-donor pairs (first acceptor then donor) are over-represented at long lag times whereas donor-acceptor pairs are underrepresented, which causes a pronounced asymmetry of the cross-correlation ratios. Similarly, donor photon pairs with long lag times are overrepresented, which causes a slow increase of $f_{DD}(\tau)$ with increasing $\tau$ paired with a decrease in $f_{AA}(\tau)$. When selecting only those bursts without acceptor photobleaching using pulsed interleaved excitation (PIE)[48–50] (Fig. 4a, b right), nearly symmetric cross-correlation ratios are retrieved. Hence, the cross-correlation ratios can be used to check whether bursts with photo-bleached acceptor dyes have been efficiently removed from the data. Notably, photo-bleaching also affects the FRET correlation functions (Fig. 4c). Whereas appropriate filtering of only those bursts without photobleached molecules retrieves the correct FRET correlation function (Fig. 4c right), the unfiltered data set causes an additional slow decay in the FRET correlation function (Fig. 4c middle), which is artificial. As a rule of thumb, an asymmetry in the two branches of the cross-correlation ratios indicates the presence of photobleaching, which can lead to a slow decay in the FRET correlation function. Hence, care must be taken when interpreting the results in such cases.

### The impact of brightness differences on FRET correlation functions

The FRET correlation function $g_E(\tau)$ is computed from raw photon traces. Experimental imperfections such as differences in the quantum yields of the dyes ($Q_A$, $Q_D$) or efficiencies of the detectors ($\xi_A$, $\xi_D$)

render $g_E(\tau)$ different from the true FRET correlation function $g_\varepsilon(\tau)$. The procedures to account for these imperfections in the calculation of correct FRET efficiencies ($\widetilde{E}$) have been discussed extensively in the past[49,51,52]. Let $n_0$ be the photon emission rate of each dye at identical excitation rates in the absence of imperfections, then $a = \xi_A Q_A n_0$ and $d = \xi_D Q_D n_0$ are the measured photon rates (brightness) of acceptor and donor in a microscope. The correction factor $\gamma = a/d$ is typically used to compute corrected FRET efficiencies. For a 2-state system with brightness differences between the dyes and ignoring antibunching, the measured and true FRET correlation functions are related by (see "Methods")

$$g_E(\tau) = \langle n \rangle^{-2} \frac{a^2 d^2}{\langle n \rangle^2 + (a - d)^2 g_\varepsilon(\tau)} g_\varepsilon(\tau) \qquad (5)$$

Here, $\langle n \rangle = a\langle\varepsilon\rangle + d(1 - \langle\varepsilon\rangle)$ is the total photon rate averaged over the conformational states. The factor in front of $g_\varepsilon(\tau)$ is time-dependent due to $g_\varepsilon(\tau)$ in the denominator, which alters the decay of $g_E(\tau)$ compared to $g_\varepsilon(\tau)$ (Fig. 4d, e). Yet, its impact is marginal because $\langle n \rangle^2$ dominates the denominator. Indeed, within the range $0.1 \le \gamma \le 10$, which by far exceeds correction factors in most smFRET experiments; the typical range in experiments is $0.5 \le \gamma \le 2$ (Fig. 4f, g); the apparent relaxation time $\tau_D$ of a 2-state system obtained from $g_E(\tau)$ does not deviate more than 6% from the true value (Fig. 4f). This deviation is further diminished in the presence of background photons. Contrary to the relaxation time, the amplitude of $g_E(\tau)$ is strongly affected by differences in the dye brightness (Fig. 4g). The timescales determined with FRET correlation functions on the other hand, are rather robust.

However, the dye brightness might be a fluctuating quantity, and two cases can be distinguished: the brightness is dependent (case 1) or independent (case 2) of the conformational states. In case 1, the result depends on the kinetic model (Fig. 4h). For a 2-state model, the FRET correlation function is given by

$$g_E(\tau) = \langle n \rangle^{-2} \frac{(n_{D,1} n_{A,2} - n_{A,1} n_{D,2})^2}{\langle n \rangle^2 (\varepsilon_1 - \varepsilon_2)^2 + (n_2 - n_1)^2 g_\varepsilon(\tau)} g_\varepsilon(\tau) \qquad (6)$$

Here, the total photon rate of state $i$ is $n_i = n_{A,i} + n_{D,i}$ with $n_{A,i} = a_i \varepsilon_i$ and $n_{D,i} = d_i(1 - \varepsilon_i)$ and the average total photon rate is $\langle n \rangle = p_1 n_1 + p_2 n_2$. Again, the leading term in the denominator ($\propto \langle n \rangle^2$) is time-independent such that $g_E(\tau)$ decays like the true correlation function $g_\varepsilon(\tau)$ (Fig. 4i). Indeed, calculations of a 2-state model show that the relaxation time differs not more than 16% from the true value even if $a_i$ and $d_i$ vary by two orders of magnitude (Fig. 4j).

If the dye brightness fluctuates independently of the structural state of the system, $g_E(\tau)$ is a combination of the photophysical and structural dynamics whose behavior depends on the specific processes that lead to the brightness fluctuations. General conclusions are difficult to draw in this case unless the fluctuations are much faster than the conformational dynamics, in which case Eq. 5 will be obtained. In summary, brightness differences of the dyes impact FRET correlation functions. Yet, within experimental limits, the error in timescale is marginal unless the brightness fluctuates independently of conformational transitions. Like model-based analysis approaches, a correlation analysis quantifies fluctuations in raw FRET efficiencies, but it does not provide the means to unambiguously identify the source of these fluctuations.

### Beyond the Poisson limit

So far, we modeled the emission of photons as a Poisson process because the nanosecond lifetimes of photophysical singlet states of the dyes are much faster than the microsecond dynamics we are interested in. However, organic fluorophores are large, conjugated π-systems that also populate triplet states with microsecond lifetimes.

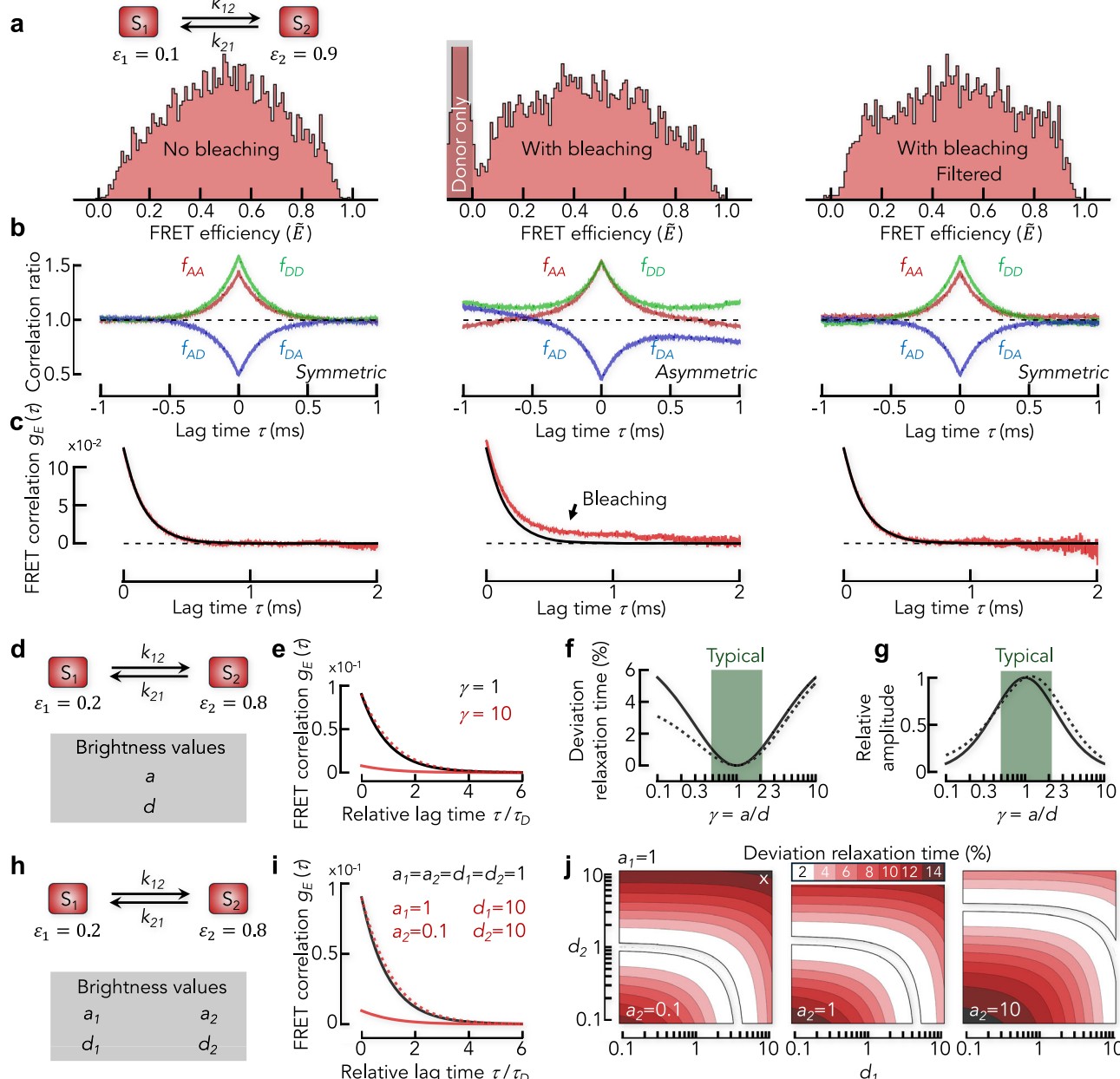

**Fig. 4 | Impact of photobleaching and dye brightness on FRET correlation functions. a** FRET efficiency histograms (corrected) for a two-state model ($k = 3\,\text{ms}^{-1}$, $\varepsilon_1 = 0.1$, $\varepsilon_2 = 0.9$) without (left) and with (middle) photobleaching. The gray shaded area indicates donor-only molecules that were excluded from the analysis ($\tilde{E} < 0$ due to acceptor direct excitation). The FRET efficiency histogram with photobleaching but filtered for unbleached bursts (right). **b** Normalized correlation ratios $f_{AA}(\tau)$ (red), $f_{DD}(\tau)$ (green), $f_{AD}(\tau)$ and $f_{DA}(\tau)$ (blue) for the data in (**a**). The correlation ratios are plotted for positive and negative lag time to better identify asymmetries of the cross-correlation ratios. **c** FRET correlation function for the data in (**a**). Black solid line is an exponential fit to the FRET correlation function in the absence of photobleaching. **d** Kinetic scheme of the 2-state model with dye brightness independent of the conformational state. **e** Analytical solution of the FRET correlation function for the 2-state model for $a = d$ ($\gamma = 1$, black) and $a = 10d$

($\gamma = 10$, red). The dashed line is the re-scaled correlation function shown for the case $\gamma = 10$. **f** Deviation of the relaxation rate from the true value without (solid black) and with background (dashed black). To highlight the difference, we used an unrealistic high background (10% of the acceptor signal). **g** The amplitude of the measured FRET correlation function relative to the value of the true amplitude is plotted as function of the correction factor. Solid and dashed lines are the same as in (**f**). **h** Kinetic scheme of the 2-state model with state-dependent dye brightness. **i** Analytical solutions of the FRET correlation function for identical brightness of all dyes and states (black line) and for a mixed case (red). The dashed red line is the rescaled correlation function shown as a solid red line. **j** Maps to indicate the relative deviation of the apparent relaxation rate (color scale) for the model in (**h**). Empty regions correspond to deviations <2%. A white cross indicates the case shown in (**i**).

Probably best characterized is the dye pair AlexaFluor488 and Alexa-Fluor594 (Fig. 5a)[53]. Nettels et al. determined the photophysical model for this pair including the transition rates (Supplementary Table 1) and showed that the FRET efficiency depends weakly on the laser power[53]. Albeit this dependence is less relevant for FRET histograms due to singlet-singlet and singlet-triplet annihilation processes[53], it might

impact the dynamics extracted from diffusion-based smFRET experiments. When a molecule diffuses through the confocal volume, excitation intensity, photon emission rates, and ultimately also the measured FRET efficiency are functions of the position in the confocal volume. To estimate the magnitude of these non-idealities, we performed Brownian dynamics simulations of molecules with a donor-

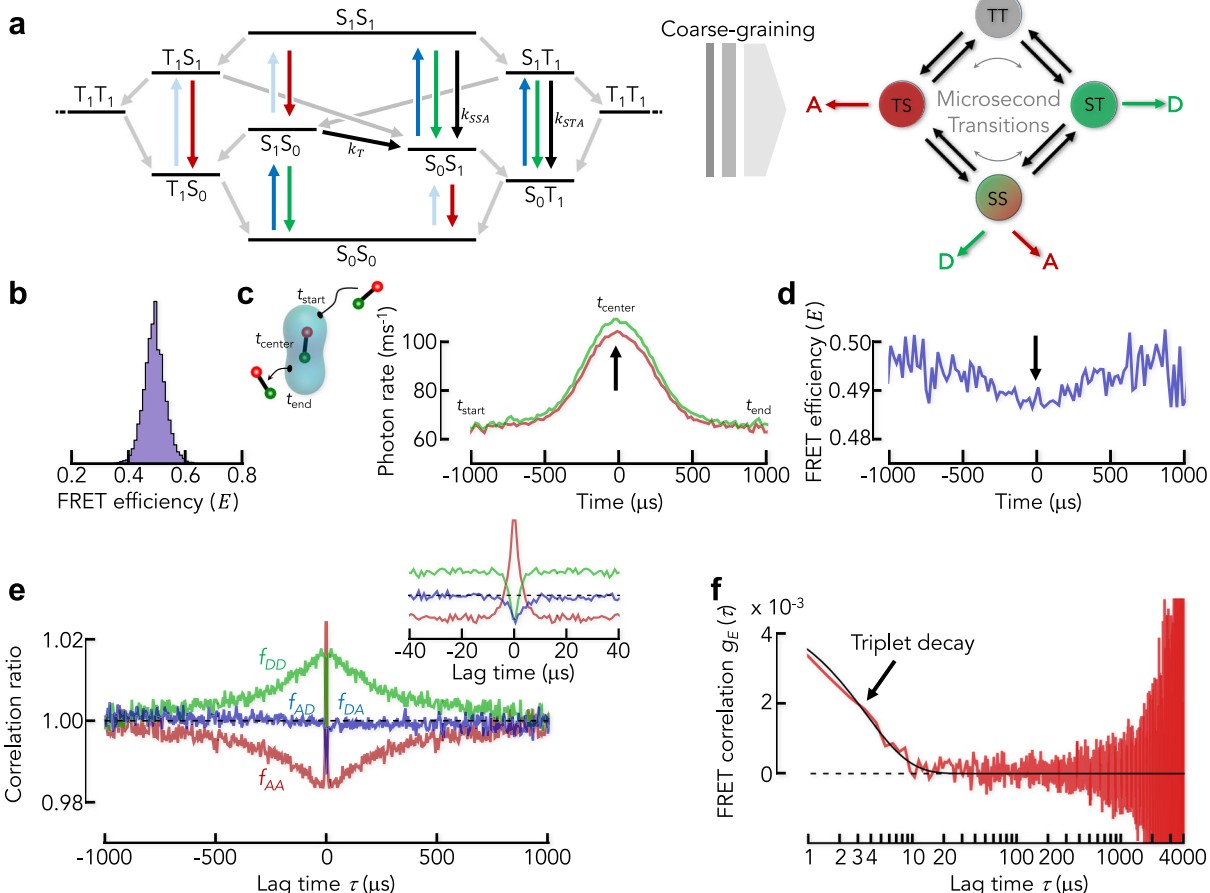

**Fig. 5 | The impact of non-Poissonian photon emission. a** Jablonski diagram of the dye pair AlexaFluor 488 and 594 based on Nettels et al. [53] (left). Each state is denoted by the electronic states of donor (first symbol) and acceptor (second symbol). S and T stand for the singlet and triplet manifolds, respectively. Subscripts refer to electronical ground (0) and first excited (1) states. Red and green arrows indicate transitions that lead to the emission of acceptor and donor photons, respectively. Dark blue arrows are excitation transitions of the donor and light blue arrows indicate the direct excitation of the acceptor at the wavelength of the donor. Black arrows indicate energy transfer processes where singlet-singlet annihilation and singlet-triplet annihilation are indicated by the rates $k_{SSA}$ and $k_{STA}$, respectively (Supplementary Table 1). The classical Förster energy transfer rate is $k_T$. Gray arrows indicate singlet-triplet and triplet-singlet transitions. The coarse-grained (CG) model of the full photophysical scheme (right) has 4 states that interconvert at microsecond timescales. **b** FRET histogram of a particle with fixed donor-acceptor distance identical to the Förster distance simulated using the coarse-grained model in (**a**). **c** Average of all photon traces of the bursts in the simulation. The overlay was constructed by aligning the trajectories relative to the average arrival time of donor (green) and acceptor (red) photons. The mean arrival time was arbitrarily set to zero. Arrow indicates the mismatch between donor and acceptor signal. **d** Apparent FRET efficiency profile calculated from the data in (**c**). **e** Normalized correlation ratios for the data in (**b**–**d**). Inset: Zoom of the normalized correlation ratios. **f** FRET correlation function for the data in (**b**–**d**). Black line is a single-exponential fit. Arrow indicates the triplet-induced decay.

acceptor distance fixed to the value of the Förster distance (Fig. 5b). Electronic transitions take place at two timescales: nanoseconds for transitions out of singlet states and microseconds for transitions out of triplet states. We take advantage of this timescale separation and coarse-grain the photophysical scheme such that the dynamics at microsecond timescales is approximately preserved in the simulation (Fig. 5a, see "Methods" and Supplementary Note 5). An overlay of the simulated photon traces based on their mean photon arrival times indeed shows a mismatch of donor and acceptor photon rates at the center of the confocal volume where the excitation intensity is highest (Fig. 5c). This results in a time-dependent FRET efficiency due to the diffusion of molecules through the confocal volume (Fig. 5d). This combination of triplet dynamics and illumination-induced FRET fluctuations strongly affects intensity correlation functions[54] and correlation ratios. We find that $f_{AA}(\tau)$ first decreases and then increases again (Fig. 5e and inset). The donor ratio $f_{DD}(\tau)$ shows the same decays, albeit with reversed amplitude signs, i.e., $f_{DD}(\tau)$ first increases due to triplet dynamics and then slowly decreases due to the apparent change

in FRET efficiency upon diffusion through the confocal volume (Fig. 5e and inset). The cross-correlation ratios ($f_{AD}(\tau)$ and $f_{DA}(\tau)$) are less affected, but they also show the fast component due to triplet dynamics whereas the slow component due to diffusion is less prominent. Notably, these components are significantly suppressed in the FRET correlation function (Fig. 5f). In fact, the slow component is virtually absent because the FRET changes when passing the excitation volume are exceedingly small (Fig. 5d). However, the fast decay resulting from the lifetime of triplet states is observed at a timescale of $3\,\mu s$. Although the triplet amplitude is predicted to be small (Fig. 5f), care should be taken in model-based analysis approaches that do not explicitly include these dye dynamics. As a complete modeling of photophysical transitions is nearly impossible given the incomplete characterization of organic fluorophores used in smFRET experiments, we recommend an empirical approach to minimize the impact of triplet dynamics. For instance, triplet dynamics can be reduced at lower laser powers, by choosing dyes with low triplet occupancy, or by using triplet-quenching additives.

## Experimental application

To demonstrate the performance of FRET correlation functions in actual experiments, we investigated the dynamics of (i) a DNA Holliday junction[19,55], (ii) the large subunit of U2AF2 from the pre-messenger RNA (mRNA) splicing machinery[56], (iii) a membrane protein proton dependent oligo-peptide transporter (POT)[57], and (iv) an intrinsically disordered protein[58].

The $Mg^{2+}$-dependent dynamics of the Holliday junction have previously been used to demonstrate the accuracy of the H²MM photon-by-photon analysis in quantifying kinetic rates from 100 to 10,000 $s^{-1}$. H²MM analysis found that a two-state model provided a reasonable description of the dynamics. We labeled a four-strand Holliday junction with AlexaFluor488 and AlexaFluor594 (Fig. 6a, Supplementary Table 3) and performed smFRET experiments using pulsed-interleaved excitation (PIE)[48], which allowed us to filter bleached molecules. At low concentration of $MgCl_2$, we found a single FRET peak centered at a corrected FRET efficiency of $\widetilde{E} = 0.45$ (Fig. 6a). When we increased the $MgCl_2$ concentration, the single peak split into two peaks at $\widetilde{E} = 0.35$ and $\widetilde{E} = 0.75$, suggesting a slowdown of the dynamics. After molecule identification and removal of photo-

bleached molecules (see "Methods"), we determined the normalized correlation ratios (Fig. 6b). The correlation ratios roughly resemble the pattern found in our simulations using the coarse-grained (CG) photophysical scheme (Fig. 5e), but different from our simulations, photophysical transitions mix with conformational transitions (Fig. 6b). The acceptor autocorrelation ratio first decreases and then increases slightly whereas the donor autocorrelation ratio decreases (Fig. 6b). The cross-correlation ratios increase monotonically and are symmetric for DA and AD photon pairs, suggesting that bursts with bleached acceptor were successfully removed. We then computed FRET correlation functions for the experiments at different $MgCl_2$ concentrations (Fig. 6c). We found two decays, a fast decay at a timescale of $\sim 15 \, \mu s$ and a slow decay with a strong dependence on the $MgCl_2$ concentration. To describe these kinetics, we fitted the FRET correlation functions with an empirical model consisting of a sum of two exponential decays and an offset. Given that the fast decay was invariant with the $MgCl_2$ concentration, we used a global fit in which the relaxation rate of the fast decay ($\lambda_1$) and the offset were global parameters, whereas the relaxation rates of the slow decay ($\lambda_2$) and the amplitudes of all decays were local parameters, i.e., specific for each $MgCl_2$

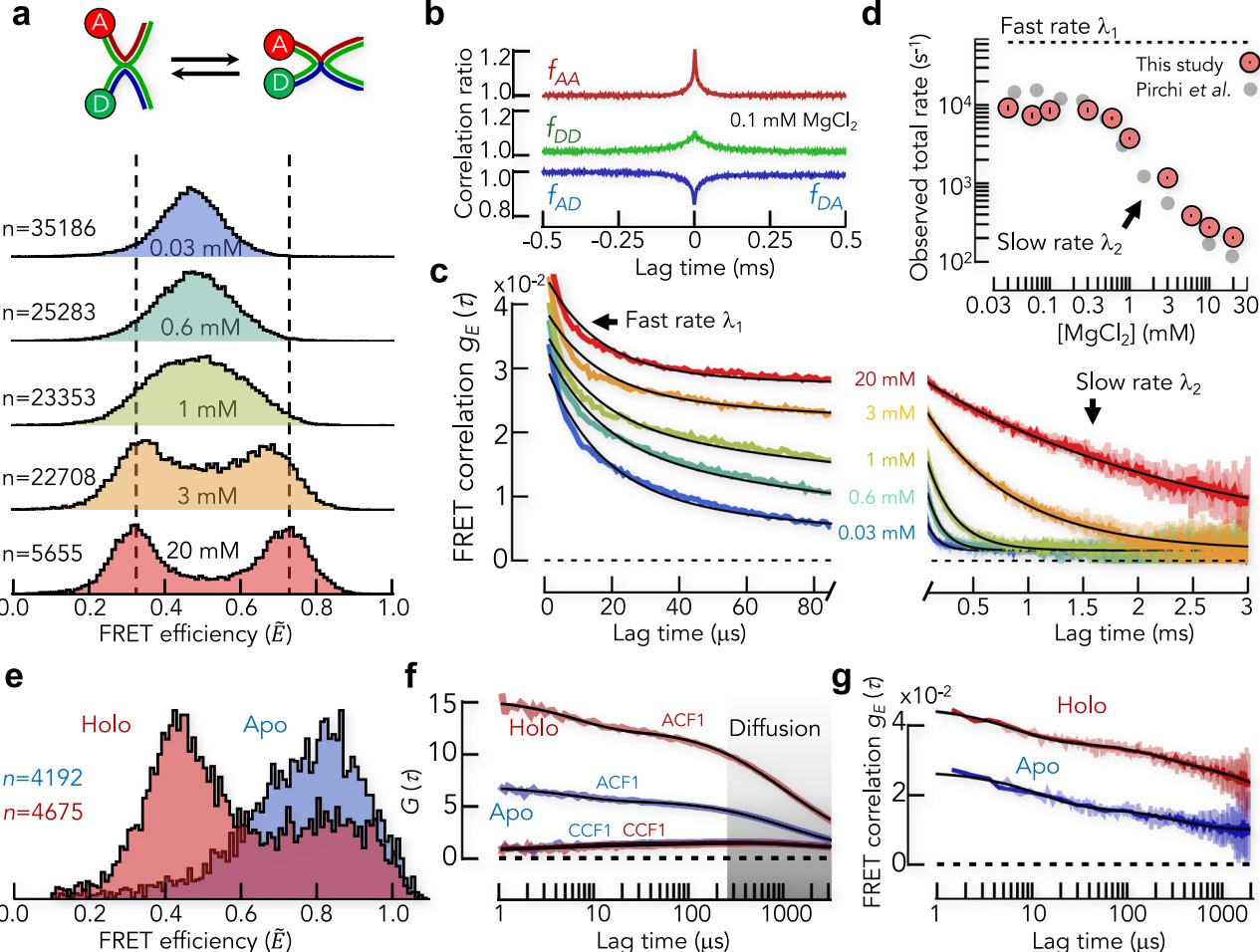

**Fig. 6 | Experimental test of FRET correlation functions. a** Scheme of the Holliday junction labeled with donor (D) and acceptor (A) switching between low and high FRET efficiency states (top). Experimental FRET efficiency histograms (measured with PIE[48] at different concentrations (indicated) of $MgCl_2$ (bottom). The number of bursts (*n*) is indicated in the histograms. **b** Correlation ratios (indicated) at 0.1 mM $MgCl_2$. **c** FRET correlation functions (color as in **a**) together with the global fit to a sum of two exponentials with an offset (black lines). The $MgCl_2$ concentrations are indicated. **d** Comparison of the kinetic rates of the fast decay (dashed line) and the slow decay (red circles) with the total rate obtained from a

previously published H²MM analysis (gray circles). Error bars represent the error from the fit. **e** FRET efficiency histograms of U2AF2 in apo- and holo-state. The number of bursts (*n*) is indicated. **f** Filtered FCS autocorrelation functions (ACF) and cross-correlation function (CCF) for apo (blue) and holo (red) state together with a fit containing two exponential decays and a diffusion component. The timescale of diffusion is indicated as gray shaded area. **g** FRET correlation functions for the data in (**e**, **f**). The black line is a fit with three exponential decays. Light colors in the FRET correlation functions in (**b**, **f**, **g**) are the data with a lag time binning of 1 μs and dark colors are for a 10 μs binning.

concentration. The fits provided a reasonable description of the FRET correlation functions (Fig. 6c). When we compared the apparent decay rates with those obtained previously with H²MM, we found an excellent agreement between the slow decay rate ($\lambda_2$) and the total rate of the two-state model used in H²MM[19] (Fig. 6d). With increasing concentration of $Mg^{2+}$ ions, the total rate decreases from $\lambda_2 = 9,515 \pm 515\,s^{-1}$ to $\lambda_2 = 212 \pm 1\,s^{-1}$. Interestingly, the fast decay for which we found a rate of $\lambda_1 = 63,360 \pm 1,370\,s^{-1}$ was also observed with H²MM and was interpreted as acceptor blinking at the time. Yet, our experiments were performed with the AlexaFluor488/594 pair instead of the Atto532/647 N pair used in the previous H²MM study (Atto532/647 N), which suggests that the fast timescale is independent of the dye-pair. In fact, the relaxation time of the fast decay $\tau_1 = 1/\lambda_1 = 15.8 \pm 0.4\,\mu s$ is substantially slower than the triplet dynamics found in our simulations of the dye pair AlexaFluor488/594 ($\sim 3\,\mu s$. Also, the amplitude of the fast decay ($0.018 \pm 0.003$) exceeds the expected amplitude from triplet dynamics ($0.004$) (Fig. 5f), which either indicates that the photophysical model of the AlexaFluor488/594 dye pair is incomplete or that extremely fast molecular motions of the DNA mix with triplet dynamics. Given that previous experiments with double-stranded DNA[29,59] identified local fluctuations at timescales of $50\,\mu s$, the fast dynamics observed in the data of the Holliday junction are likely a mixture of triplet (blinking) dynamics of the dyes and fast local fluctuations of the DNA. In summary, these experiments show that FRET correlation functions reproduce the results from a photon-by-photon analysis quantitatively, which demonstrates the reliability of this method in extracting dynamics from diffusion-based smFRET experiments.

Next, we compared the results from FRET correlation functions to those of another popular model-free analysis method of dynamics, lifetime-fFCS[24,25]. To this end, we used data on the protein U2AF2 from the pre-mRNA splicing machinery, which had previously been published in a smFRET benchmark study[56]. Here, two folded domains are connected by a flexible linker such that the protein fluctuates between domain-bound (closed) and detached (open) conformations. The protein was labeled with Atto532 as donor and Atto643 as acceptor dye using the variant L187C/G326C. In the closed apo-state, the FRET efficiency is high but binding of a ligand (5 mM U9-RNA) shifts the ensemble to the open holo-state with a lower FRET efficiency (Fig. 6e). Previous fFCS-experiments[56] identified dynamics at two timescales, $\tau_1 = 9 \pm 3\,\mu s$ and $\tau_2 = 300 \pm 90\,\mu s$ (Fig. 6f). We used the raw data from laboratory #2 in the benchmark study[56] and computed FRET correlation functions for the protein in apo- and holo-state. The correlation functions show a complex behavior and at least three exponential decays were required for fitting. We found $\tau_1 = 12 \pm 6\,\mu s$ and $\tau_2 = 356 \pm 54\,\mu s$ for the apo-state together with $\tau_1 = 6 \pm 4\,\mu s$ and $\tau_2 = 230 \pm 50\,\mu s$ for the holo-state, which demonstrates an excellent agreement with the fFCS-analysis. The third decay ($\tau_3$) was extremely long (>3 ms), leads to a substantial offset and is therefore better interpreted as static heterogeneity or dynamics substantially slower than the diffusion time $t_D$. Notably, due to the significant impact from diffusion in the fFCS-analysis (Fig. 6f), information on static heterogeneity is lost whereas this information is easily accessible with FRET correlation functions.

As a final benchmark, we compare the FRET correlation functions with another model-free method, RASP[2,30]. To this end, we study the bacterial di- and tripeptide permease DtpA (E. coli) (Fig. 7a, inset) that uses a proton gradient to transport di- and tri-peptides across the membrane[57]. The two helix bundles of DtpA have been proposed to open and close in an alternating fashion. Using RASP[30], we showed previously that the distance between the helix bundles on the cytoplasmic side fluctuates at a timescale of ~1 ms when the protein is embedded in LMNG micelles[8]. To confirm these dynamics, we used the DtpA-variant W203C/Q487C with donor (AlexaFluor488) and acceptor (AlexaFluor594) being attached to the helix bundles at the cytoplasmic

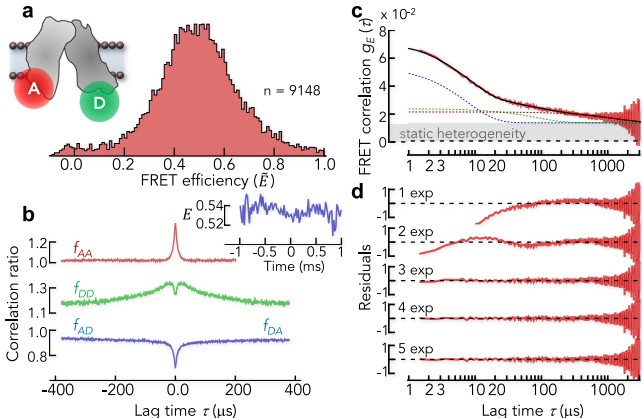

**Fig. 7 | Dynamics of the membrane protein DtpA. a** FRET efficiency histogram of DtpA in the detergent LMNG labeled at the cytoplasmic side of the two domains (inset). **b** Normalized correlation ratios of DtpA and averaged FRET trajectory of all bursts (inset). **c** FRET correlation function of DtpA computed from the data in (**a**). Solid black line is a fit with a sum of three exponential decays. Dashed blue line is the decay of the fastest component, green dashed line is the decay of the intermediate component, and red dashed line is the decay of the slowest component. The gray shaded area indicates the static heterogeneity (offset). **d** Residuals of the fits with multiple exponential decays. The number of exponentials is indicated.

side (Fig. 7a, inset). The FRET histogram of the transporter shows a broad distribution that indicates a heterogeneous mixture of different conformational states (Fig. 7a). Again, the correlation ratios resemble the pattern found in our simulations using the CG photophysical scheme (Fig. 5e). The cross-correlation ratios are symmetric (Fig. 7b), indicating a successful filtering of photobleached molecules. The FRET correlation function shows a complex decay with multiple components and an offset (Fig. 7c). We empirically fitted the data with a sum of up to five exponential decays and a manually determined offset to account for static heterogeneity (Fig. 7d). Three exponential decays are sufficient to describe the data set. The relaxation times are $\tau_1 = 8 \pm 1\,\mu s$, $\tau_2 = 86 \pm 15\,\mu s$, and $\tau_3 = 1373 \pm 68\,\mu s^{-1}$. Although the fastest timescale is close to the dynamics of photophysical triplet states (Figs. 7c and 5f), it is still more than twofold slower than expected for triplet dynamics. Like for the Holliday junction, the amplitude of this fast decay is substantially higher than in our simulations of the AlexaFluor488/594 dye pair (Fig. 5f), indicating that protein motions mix with triplet dynamics. Importantly, the slower processes are within the range found with RASP. In fact, a weighted average of the slow relaxation times gives $1.2 \pm 0.1$ ms, in good accord with the value found with RASP ($\sim 1$ ms)[8]. Also, the offset indicates the presence of static heterogeneity in accord with the previous findings with RASP[8]. Hence, we found a good agreement between the results of FRET correlation functions and the established procedure of RASP. Different from RASP, however, which requires many hours of data acquisition and tens of thousands of bursts, the FRET correlation functions computed here provide excellent signal-to-noise with only a few thousand bursts.

Finally, we demonstrate that FRET correlation functions are not restricted to microsecond timescales. We investigated the IDP ΔMyc, which is a modified version of the DNA-binding domain of the transcription factor c-Myc. In this modification, all hydrophobic residues were replaced by serine and glycine residues[58], which ensures that this sequence is largely unstructured. We recently studied the reconfiguration dynamics of ΔMyc[35] and found fast sub-microsecond timescales as expected for an IDP[11,32,33,42,60-62]. We therefore do not expect dynamics at timescales of microseconds. We performed sub-population specific nsFCS experiments[42] with ΔMyc at a denaturant concentration of 2 M GdmCl to exclude any residual transient structure formation. We collected >800,000 bursts to also probe the nanosecond dynamics with sufficient photon pair statistics (Fig. 8a,

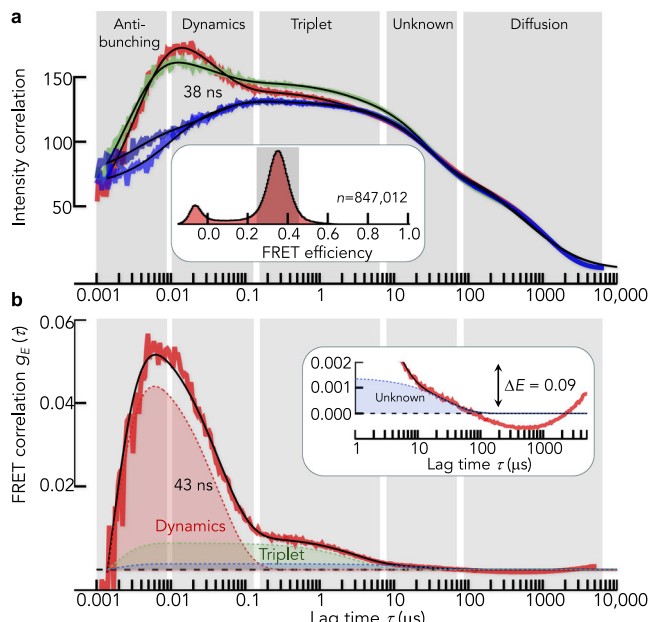

**Fig. 8 | Dynamics of the IDP ΔMyc. a** Acceptor (red) and donor (green) intensity autocorrelation functions and cross-correlation functions (blue) for ΔMyc. The FRET efficiency histogram, together with the range of bursts chosen for the calculation of the correlation functions, is shown as an inset. Vertical gray bars indicate the decay components. Black lines are global fits with 5 decay components (see "Methods"). **b** FRET correlation function computed for the data in (**a**). The black line is a fit with 4 decays (see "Methods"). Inset: zoom at the long-time microsecond regime.

inset). The classical intensity correlation functions show five decay components: anti-bunching, conformational dynamics, triplet dynamics, an unknown component, and the decay due to the diffusion of molecules through the confocal volume (Fig. 8a). Conformational dynamics due to chain reconfigurations differ from other decays by a rise in the cross-correlation functions. Compared to the complexity of the intensity correlation functions, the FRET correlation function is simple (Fig. 8b). Only antibunching, reconfiguration, and triplet dynamics are left with significant amplitude, whereas the dominant diffusion component is efficiently suppressed. The reconfiguration timescales are 38 ns (intensity correlation) and 43 ns (FRET correlation), i.e., very similar. The slower timescale of triplet dynamics agrees well with our simulations of the photophysical scheme, both in terms of timescale (3 $\mu$s) and amplitude (0.006) (see Fig. 5f for comparison). The unassigned decay at 27 $\mu$s that is prominent in the intensity correlation functions is also substantially suppressed in the FRET correlation function, indicating that it does not result from changes in FRET efficiency. Finally, due to the lack of dynamics at timescales >30 $\mu$s, the data provide a chance to check the quality of diffusion suppression in FRET correlations. A zoom at microsecond timescales shows residual fluctuations with an amplitude of 0.002 (3% of the total amplitude). This value can be considered as a lower limit, i.e., the amplitude of decays in FRET correlation functions should be >0.002 to be considered significant. We can translate this amplitude into a lower detection limit for FRET changes. Assuming a 2-state model, the apparent FRET efficiencies $E_1$ and $E_2$ of the two conformational states should differ by at least 0.09 FRET efficiency units to overcome the residual diffusion artifacts in FRET correlation functions, i.e., to exceed an amplitude of 0.002.

## Discussion
SmFRET experiments have become an integral part of a "new" era in structural biology that aims at establishing a holistic picture of proteins as dynamic structures[63,64]. The theoretical foundations to

rigorously analyze smFRET experiments have been laid over the past two decades[15,44,65–72]. Here, we presented a helpful addition. We showed how the conformational dynamics of single molecules can be retrieved from FRET correlation functions in diffusion-based smFRET experiments. The advantages of the method are fourfold. First, it is simple and can be applied to any smFRET experiment with single-photon detection, irrespective of the data acquisition mode (pulsed- or continuous wave excitation, 2-channel or 4-channel detection). Second, the method effectively suppresses the diffusion component in standard correlation functions, thus providing direct insights into dynamics at timescales from micro- to milliseconds. Third, a few thousand molecules are sufficient to obtain a reasonable signal-to-noise ratio in FRET correlation functions, which substantially reduces measurement times compared to other model-free approaches such as RASP[30], 2D-FLCS[27,28], and lifetime-fFCS[24,25]. Fourth, the method provides diagnostic tools to identify static heterogeneity, artifacts such as photobleaching, and it serves as a rigorous test for model-based fitting approaches that are standardly used in diffusion-based smFRET and integrative structural biology approaches nowadays[9,10,15,19,20,64,73,74]. Hidden-Markov-model fitting of smFRET data with diffusing molecules suffers from our incomplete knowledge on the impact of diffusion through the inhomogeneously illuminated confocal volume and photophysical non-idealities such as triplet dynamics on the photon traces. Are dynamics at tens or hundreds of microseconds poised by the fluctuating photon rate due to triplet and diffusion? Our results demonstrate that, albeit these non-idealities can never completely be eradicated, they are substantially suppressed on the FRET efficiency coordinate, which implicitly supports the validity of Markov-model fitting approaches in diffusion-based smFRET experiments. Most importantly, FRET correlation functions provide a model-independent tool to extract dynamic timescales from the photon traces of individual molecules. This is of particular importance for benchmarking Hidden–Markov models.

In conclusion, our tests of the method explored the impact of protein concentrations, photobleaching, quenching, triplet, photophysical saturation, and different detection volumes. Two processes impact FRET correlation functions most: (i) photobleaching, which introduces slow decays in the correlation functions, and (ii) triplet dynamics, which causes fast decays at timescales 1–10 $\mu$s for the Alexa dye-pair used here. With the calculation of cross-correlation ratios, we provide the diagnostic tool to identify photobleaching. Whereas experiments performed with PIE, ALEX[48–50] are ideal to remove bleached molecules, the fluctuations due to triplet transitions are unavoidable. We hope that the simple model-free FRET correlation approach presented here will be a useful addition to the current toolkit of smFRET experiments.

## Methods
### Data simulations and numerical calculations
**Simulation of Brownian diffusion.** To test the performance of our method, we performed simulations of the photon time traces of diffusing molecules that undergo structural dynamics while diffusing through a confocal spot using the Fretica package (https://schuler. bioc.uzh.ch/programs/), developed by Daniel Nettels and Benjamin Schuler (University of Zurich). We obtained trajectories of diffusing particles with Brownian dynamics in spherical coordinates, assuming a radially symmetric confocal volume, which is located at the origin of the coordinate system. The time evolution of particle radial coordinates $r(t)$ was obtained with

$$r(t + \Delta t) = r(t) + \frac{2D\Delta t}{r(t)} + \Delta r \qquad (7)$$

where $\Delta t$ is a timestep, $D$ is the diffusion coefficient, and $\Delta r$ is a stochastic move drawn from a normal distribution with zero mean and

variance $\sigma_{\Delta r}^2 = 2D\Delta t$. We used $\Delta t = 1\,\mu s$, except for simulations with the CG photophysical model (Fig. 5a), for which we used $\Delta t = 0.5\,\mu s$. The diffusion coefficient of the particles was $D = 50\,\mu m^2/s$, which corresponds to a medium-sized protein (Stokes radius of 4.3 nm at room temperature in water). The simulations were initialized by randomly placing particles in a simulation sphere with a radius of $R = 3\,\mu m$, afterwards each particle was simulated until it leaves this sphere. The number of initial particles was drawn from a Poisson distribution with a mean $n_0 = 4\pi R^3 c_0/3$, where $c_0$ is the bulk particle concentration. The loss of particles that diffuse out of the simulation sphere was compensated by periodically (with period $T_{new} = 1000\Delta t$) placing new particles inside the sphere. The number of inserted particles and the distribution of their positions $c_{new}(r)$ were chosen based on the solution of the diffusion equation that describes how an empty sphere fills over time $T_{new}$ due to the constant bulk concentration at its outer border

$$\frac{\partial c}{\partial t} = D\left(\frac{\partial^2 c}{\partial r^2} + \frac{2}{r}\frac{\partial c}{\partial r}\right) \tag{8}$$

with the initial conditions $c(r < R, t = 0) = 0$ and the boundary conditions $c(r = R, t) = c_0$ and $c(r \to 0, t) = 0$. The solution of Eq. 8 is known[75] and is given by

$$\frac{c(r,t)}{c_0} = 1 + \frac{2R}{\pi r}\sum_{n=1}^{\infty}\frac{(-1)^n}{n}\sin\left(\frac{n\pi r}{R}\right)\exp\left(-Dn^2\pi^2 t/R^2\right) \tag{9}$$
$$\text{and } c_{new}(r) = c(r, T_{new})$$

The mean number of particles that enter the sphere $n_{new}$ was calculated by integrating $c_{new}(r)$ over the volume of the sphere. After each time interval $T_{new}$, a random number of new particles was drawn from the Poisson distribution with mean $n_{new}$ and the positions inside the sphere were randomly chosen from the distribution with the density function $P_{new}(r) = 4\pi r^2 c_{new}(r)/n_{new}$. In total, we simulated particle trajectories for 1800 s in most cases whereas 3600 s long trajectories were used for simulations of a triplet kinetics model.

**Simulation of conformational dynamics and photon traces.** Once the particle trajectories were simulated, we added stochastic conformational dynamics simulated according to the rate equation

$$\frac{d\mathbf{p}}{dt} = \mathbf{K}\mathbf{p} \tag{10}$$

where $\mathbf{p}$ is the population vector of states and $\mathbf{K}$ is a matrix that contains the transition rates between states (see below). Each state is characterized by the rate of acceptor and donor photon emission, which is proportional to the intensity profile in the confocal spot

$$I(r) = \exp\left(-\frac{2r^2}{w_0^2}\right)\text{ with } w_0 = 0.4\,\mu m \tag{11}$$

such that the total photon rate at the center of the excitation volume is $\lambda_{tot} = 0.4$ MHz. Given the trajectories of positions and conformational states, we simulated the emission of photons. Unless stated otherwise, we used a realistic background photon rate of $\lambda_d = 2$ kHz for the donor channel and $\lambda_a = 1$ kHz for the acceptor channel. Except for the simulation of non-Markov dynamics (Fig. 3) and in simulations of the CG photophysical model (Fig. 5a), we introduced different detection efficiencies for the dyes ($\gamma = Q_a\xi_a/Q_d\xi_d = 1.15$), where $Q_{a,d}$ and $\xi_{a,d}$ are the quantum yields and detection efficiencies for acceptor and donor dye, respectively. Crosstalk (leakage) of donor photons into the acceptor channel ($\beta_{DA} = 0.05$) and of acceptor photons into the donor channel ($\beta_{AD} = 0.003$) together with the probability to directly excite the

acceptor with the donor excitation laser ($\alpha = 0.048$) were also included.

**Two state model.** In our simulations of the 2-state model (Fig. 2), we simulated particles that switch between a low FRET-state 1 and a high FRET-state 2. The two states are characterized by the true FRET efficiencies $\varepsilon_1 = 0.1$ and $\varepsilon_2 = 0.9$. The kinetic rate matrix for this model is

$$\mathbf{K} = \begin{pmatrix} -k & k \\ k & -k \end{pmatrix} \tag{12}$$

For simplicity, forward and backward rates were equal $k$. We simulated the systems with wide range of values for $k$ (0.1, 0.08, 0.05, 0.03, 0.025, 0.015, 0.01, 0.008, 0.005, 0.003, 0.002, 0.001, 0.0008, 0.0005, 0.0003, 0.0002, 0.00015, 0.0001, 0.00005) given in units of $\mu s^{-1}$. We also simulated the system with $k = 0\,\mu s^{-1}$ to access static heterogeneity. The initial state for each particle was chosen randomly with equal probabilities to be in low or high FRET state. All the above simulation were done for particle trajectories obtained with different bulk concentrations (25.0, 50.0, 100.0, 200.0, 300.0, 400.0, 500.0), given in units of pM. A threshold of 100 photons was used to identify bursts. We also provide examples of simulations with unequal forward and backward rates in Supplementary Fig. 5 and simulations that explore the effect of spectral crosstalk (leakage of donor emission into the acceptor channel) in Supplementary Fig. 6.

**Non-Markov model and likelihood maximization.** Simulations of the photon traces for the 2-state Markov and 4-state non-Markov model were performed as described above. We simulated "pure" systems in the absence of background and instrumental imperfections ($\gamma = 1$, $\beta_{DA} = \beta_{AD} = 0$, $\alpha = 0$) such that $E = \widetilde{E}$ and identified bursts using a threshold of 50 photons. The rate matrix of the 2-state system is given by Eq. 12 and the rate matrix of the 4-state non-Markov system is given by

$$\mathbf{K} = \begin{pmatrix} -k & k & 0 & 0 \\ k & -2k & k & 0 \\ 0 & k & -2k & k \\ 0 & 0 & k & -k \end{pmatrix} \tag{13}$$

The fit of the simulated photon traces with the 2-state (Eq. 12) and 4-state (Eq. 13) models was done by maximizing the log-likelihood function of all photon trajectories simultaneously. The log-likelihood function for the i's burst with $N$ photons is given by

$$L_i = \ln\left(\mathbf{1}^{\mathrm{T}}\left(\prod_{j=2}^{N}\mathbf{F}(c_{ji})e^{-(\mathbf{K}-n\mathbf{I})\tau_{ji}}\right)\mathbf{F}(c_{1i})\mathbf{p}_{eq}\right)\text{ with }\mathbf{F}(c) = \begin{cases} \mathbf{V}_A & c = A \\ \mathbf{V}_D & c = D \end{cases} \tag{14}$$

Here, $\mathbf{p}_{eq}$ is the vector of equilibrium probabilities that is the solution of Eq. 10 with $d\mathbf{p}/dt = \mathbf{0}$ where $\mathbf{0}$ and $\mathbf{1}$ are vectors of zeros and ones, respectively, the superscript T indicates the transposed vector, $\tau_{ji}$ is the time between photon $j$ and $j-1$ in the $i^{th}$ burst, $\mathbf{I}$ is the unit matrix, $\mathbf{V}_A$ and $\mathbf{V}_D$ are the detection matrices for acceptor and donor, respectively. For the 2-state model, the detection matrices are

$$\mathbf{V}_A = n\begin{pmatrix} \varepsilon_1 & 0 \\ 0 & \varepsilon_2 \end{pmatrix}\text{ and }\mathbf{V}_D = n\begin{pmatrix} 1-\varepsilon_1 & 0 \\ 0 & 1-\varepsilon_2 \end{pmatrix} \tag{15}$$

Here, $n = n_A + n_D$ is the total photon emission rate of donor and acceptor with $n_A = \mathbf{1}^{\mathrm{T}}\mathbf{V}_A$ and $n_D = \mathbf{1}^{\mathrm{T}}\mathbf{V}_D$. We maximized the log-likelihood of all $M$ photon traces given by $\mathcal{L} = \sum_{i=1}^{M}L_i$ with respect to the model parameters $k$, $\varepsilon_1$, $\varepsilon_2$, and $n$ using the Fretica software package. To compute the expected FRET correlation functions, we generated synthetic data from the measured photon trajectories using a

recoloring approach, where new "colors" (donor or acceptor) for the photons are simulated while their detection times are maintained unchanged. The re-coloring procedure[67] includes two steps: (1) Based on the rate coefficient, $k$, obtained from the likelihood maximization, we simulate a time-trajectory of conformational states for the given sequence of photon detection times of a burst. (2) New photon "colors" (donor or acceptor) are chosen randomly for each photon according to the $\varepsilon_i$ values attributed to the simulated conformations. Finally, FRET histograms and FRET correlation functions are calculated from the recolored bursts. To minimize noise, we recolored the data ten times and computed average FRET histograms and correlation functions. We also computed the theoretical FRET correlation function of the 4-state model using

$$g_E(\tau) = g_\varepsilon(\tau) = \mathbf{1}^T \varepsilon e^{\mathbf{K}\tau} \varepsilon \mathbf{p}_{eq} - \left( \mathbf{1}^T \varepsilon \mathbf{p}_{eq} \right)^2 \qquad (16)$$

with

$$\varepsilon = \begin{pmatrix} \varepsilon_1 & 0 & 0 & 0 \\ 0 & \varepsilon_1 & 0 & 0 \\ 0 & 0 & \varepsilon_2 & 0 \\ 0 & 0 & 0 & \varepsilon_2 \end{pmatrix} \qquad (17)$$

The function is shown as dashed line in Fig. 3d. Notably, Eq. 16 is only correct under ideal conditions (no differences in dye brightness, see Supplementary Notes 1 and 2), which are fulfilled in this simulation. Under non-ideal conditions, Eq. 2 must be used to compute the analytical FRET correlation function (see Eqs. 19–22).

**Simulations with photobleaching.** To access the effect of photobleaching (Fig. 4a–c), we modelled a system with four states: donor-only ($D$), acceptor-only ($A$), low FRET ($DA_1$) with $\varepsilon_1 = 0.1$, and high FRET ($DA_2$) with $\varepsilon_2 = 0.9$. The rate matrix $\mathbf{K}$ in this case is a combination of the rate matrix $\mathbf{K}_0$ for conformational transitions between $DA_1$ and $DA_2$ and the rate matrix $\mathbf{K}_1$ that describes photobleaching. Since photobleaching is a function of the excitation intensity, the total rate matrix involves the excitation profile (Eq. 11) and is now given by

$$\mathbf{K} = \mathbf{K}_0 + I(r)\mathbf{K}_1 \qquad (18)$$

With the matrices

$$\mathbf{K}_0 = \begin{pmatrix} 0 & 0 & 0 & 0 \\ 0 & -k & k & 0 \\ 0 & k & -k & 0 \\ 0 & 0 & 0 & 0 \end{pmatrix} \text{ and } \mathbf{K}_1 = \begin{pmatrix} 0 & k_A\varepsilon_1 & k_A\varepsilon_2 & 0 \\ 0 & -k_A\varepsilon_1 - k_D\hat{\varepsilon}_2 & 0 & 0 \\ 0 & 0 & -k_A\varepsilon_2 - k_D\hat{\varepsilon}_2 & 0 \\ 0 & k_D\hat{\varepsilon}_1 & k_D\hat{\varepsilon}_2 & 0 \end{pmatrix}$$

where $\hat{\varepsilon}_i \equiv 1 - \varepsilon_i$. Here, $k_A$ and $k_D$ are acceptor and donor photobleaching rates at the center of the confocal volume ($r = 0$), respectively. We used realistic photobleaching rates of $k_A = k_D = 8 \times 10^{-4}\,\mu s^{-1}$. The conformational rates were $k = 3 \times 10^{-3}\,\mu s^{-1}$. The initial state was chosen randomly according to the probability vector $\mathbf{p}_0 = (0.1 \quad 0.4 \quad 0.4 \quad 0.1)^T$ in the basis $\{D, DA_1, DA_2, A\}$. We also simulated photon emission using a PIE scheme[48–50] of both dyes with $\gamma_{PIE} = 2$ (defined in Eq. 26) to allow the filtering of bleached molecules in the analysis. To this end, we simulated photon emissions after both donor- and acceptor excitation and experimental instrumental response functions (IRF) were used to get realistic arrival time distributions within each PIE period. A threshold of 75 photons was used to identify bursts.

**Simulation of the coarse-grained photophysical scheme.** To understand how the photophysics of real dye pairs impact the photon trajectories of molecules that diffuse through an inhomogeneously illuminated confocal volume, we developed the CG model of the kinetic scheme (Fig. 5a) published by Nettels et al. (Supplementary Note 5)[53]. The original model includes ground singlet states ($S_0$), excited singlet states ($S_1$) and triplet states ($T_1$) for both dyes, resulting in the 9 possible states of the system: $\{S_0S_0, S_1S_0, S_0S_1, S_1S_1, T_1S_0, T_1S_1, S_0T_1, S_1T_1, T_1T_1\}$. The first and second letter denote the state of donor and acceptor, respectively. Upon absorption of a photon, the dyes undergo transitions from ground to excited singlet states ($S_0 \to S_1$) with the rates $k_{ex}$ and $\alpha k_{ex}$ for donor and acceptor, respectively, where α is the probability to directly excite the acceptor at the excitation wavelength of the donor. The value of $k_{ex}$ is proportional to the intensity of the incident light and fluctuates due to the diffusion of the particle though the confocal volume. In addition to the classical relaxation pathways $S_1 \to S_0$, non-radiative interstate crossings $S_1 \to T_1$ and $T_1 \to S_0$ and singlet-singlet and singlet-triplet annihilation routes must be considered (Fig. 5a). The timescale of radiative transitions is known to be nanoseconds[53] whereas intersystem crossings occur at microsecond timescales. In addition, fluctuations of $k_{ex}$ due to diffusion are in the order of tens to hundreds of microseconds. Considering this separation of timescales, we built the CG model by assembling states into four groups: $SS = \{S_0S_0, S_1S_0, S_0S_1, S_1S_1\}$, $TS = \{T_1S_0, T_1S_1\}$, $ST = \{S_1T_0, S_1T_1\}$, and $TT = \{T_1T_1\}$ (Fig. 5a, Supplementary Note 5). Notably, transitions between these four groups (CG states) corresponds to non-radiative intersystem crossings in the original model. Radiative nanosecond transitions are modelled in form of donor and acceptor photon rates for each CG state, leading to the emission rates $\left\{ \tilde{n}_{SS}^A, \tilde{n}_{TS}^A, \tilde{n}_{ST}^A, \tilde{n}_{TT}^A \right\}$ and $\left\{ \tilde{n}_{SS}^D, \tilde{n}_{TS}^D, \tilde{n}_{ST}^D, \tilde{n}_{TT}^D \right\}$ with $\tilde{n}_{TT}^A = \tilde{n}_{TT}^D = 0$. Transition rates between CG-states were obtained by equating the steady-state fluxes between CG states with the sum of fluxes of all corresponding transitions of the full-model (Supplementary Fig. 2). In the realistic range $k_{ex} = 0.0 - 0.08\,ns^{-1}$, we found an approximately linear dependence of the emission rates on $k_{ex}$ (Supplementary Fig. 3). Generally, one would expect that also the transition rates of the CG model depend on $k_{ex}$. We found that $S \to T$ transitions, such as $SS \to TS$ or $ST \to TT$, depend nearly linearly on $k_{ex}$, whereas $T \to S$ transitions, such as $TS \to SS$ or $TT \to TS$, are insensitive to changes in $k_{ex}$ (Supplementary Fig. 2). This allowed us to write the kinetic rates matrix of the CG model in form of Eq. 18. Here, $\tilde{\mathbf{K}}_0$ contained all $T \to S$ transition rates whereas $\tilde{\mathbf{K}}_1$ contained the $S \to T$ transitions that were linearly dependent on $k_{ex}$. The dependence of the excitation rate on the location in the confocal volume is given by Eq. 11 with $k_{ex} = 0.08$ ns$^{-1}$ in the center of simulated confocal spot ($r = 0, I(0) = 1$). The initial state of each simulated particle was $SS$ as it is the equilibrium state at zero intensity outside the confocal spot. The photon rates of the CG states were obtained from a fit of the CG model to numerical results of the full model and were uniformly adjusted such that the maximum photon rate in the center of the confocal volume is $\lambda_{tot} = 0.4\,\mu s^{-1}$, which results in a realistic detection rate of 160 photons per millisecond. All parameters of the full and CG model are reported in Supplementary Tables 1–2. A direct comparison between the 9-state model and the CG model is shown in Supplementary Fig. 4.

**Calculations of FRET correlation functions with different brightness.** For the calculation of Eqs. 5–6 and the numerical results shown in Fig. 4d–j, we used the framework developed by Gopich & Szabo[16]. For a 2-state model, the detection matrices for acceptor and donor are defined by

$$\mathbf{V}_A = \mathbf{A}\varepsilon + \mathbf{A}_b \text{ and } \mathbf{V}_D = \mathbf{D}(\mathbf{I} - \varepsilon) + \mathbf{D}_b \qquad (19)$$

with the matrices

$$\mathbf{A} = \begin{pmatrix} a_1 & 0 \\ 0 & a_2 \end{pmatrix}, \mathbf{D} = \begin{pmatrix} d_1 & 0 \\ 0 & d_2 \end{pmatrix}, \varepsilon = \begin{pmatrix} \varepsilon_1 & 0 \\ 0 & \varepsilon_2 \end{pmatrix}, \mathbf{A}_b = \begin{pmatrix} b_A & 0 \\ 0 & b_A \end{pmatrix}, \mathbf{D}_b = \begin{pmatrix} b_D & 0 \\ 0 & b_D \end{pmatrix}$$

Here, $\varepsilon$ is the matrix of true FRET efficiencies, $\mathbf{A}$ and $\mathbf{D}$ are the brightness matrices, containing the brightness values of the states, and $\mathbf{A}_b$ and $\mathbf{D}_b$ contain the background photon counts in the acceptor $b_A$ and donor $b_D$ channel. We use Eq. 2 to compute the FRET correlation function in this case. To this end, we first compute the photon-pair correlation functions between photons of type X and Y according to

$$g_{XY}(\tau) = \mathbf{1}^{\mathrm{T}} \mathbf{V}_X e^{\mathbf{K}\tau} \mathbf{V}_Y \mathbf{p}_{eq} \tag{20}$$

Here, $\mathbf{K}$ is the rate matrix of the 2-state model (Eq. 12) and $\mathbf{p}_{eq}$ is the equilibrium vector of states. The correlation function of all pairs irrespective of color is

$$g(\tau) = \mathbf{1}^{\mathrm{T}} (\mathbf{V}_A + \mathbf{V}_D) e^{\mathbf{K}\tau} (\mathbf{V}_A + \mathbf{V}_D) \mathbf{p}_{eq} \tag{21}$$

Since $g_{XY}(\tau) \propto N_{XY}(\tau)$ and $g(\tau) \propto N(\tau)$ with $\frac{g_{XY}(\tau)}{g(\tau)} = \frac{N_{XY}(\tau)}{N(\tau)}$, we can use Eq. 2 together with the definition of the uncorrected mean FRET efficiency

$$\langle E \rangle = \frac{\mathbf{1}^{\mathrm{T}} \mathbf{V}_A \mathbf{p}_{eq}}{\mathbf{1}^{\mathrm{T}} \mathbf{V}_A \mathbf{p}_{eq} + \mathbf{1}^{\mathrm{T}} \mathbf{V}_D \mathbf{p}_{eq}} \tag{22}$$

to compute $g_E(\tau)$. For the special case of state-independent brightness, we set $a_1 = a_2 = a$ and $d_1 = d_2 = d$ in Eq. 19.

**Fitting of correlation functions.** The experimental FRET correlation functions were fitted with a sum of exponentials and an offset. The offset was determined either by fitting or based on the final 100 $\mu$s of the correlation function. The correlation functions of the Holliday junction (Fig. 6c) were fitted globally. The global parameters for each $MgCl_2$ concentration were the fast relaxation time $\tau_1$ and the offset. All amplitudes together with the slow relaxation time $\tau_2$ were local parameters. FRET correlation functions from Brownian dynamics simulations with Poisson photon emission statistics (Fig. 2c, e and Fig. 3b, d) were fitted with a single exponential function without offset. Given that these decays only show a single decay component, we fitted the data, including relative weights for each lag time that were given by the number of total photon pairs at this lag time.

The intensity correlation functions of the nsFCS experiments of $\Delta$Myc (Fig. 8a) were globally fitted with

$$G_{XY} = 1 + a_{XY} \frac{(1 - c_{ab}e^{-t/\tau_{ab}})(1 \pm c_b e^{-t/\tau_b})(1 + c_T e^{-t/\tau_T})(1 + c_S e^{-t/\tau_S})}{\left(1 + \frac{t}{t_D}\right)\sqrt{1 + s^2 \frac{t}{t_D}}} \tag{23}$$

Here, $X$ and $Y$ indicate the photon pairs $(AA, DD, AD, DA)$, $a_{XY}$ is the inverse average number of molecules in the confocal volume, and the indices $ab$, $b$, $T$, $S$ indicate the amplitudes $c$ and relaxation times $\tau$ of antibunching, bunching (conformational dynamics), triplet, and unassigned process, respectively. The denominator describes the diffusion where $t_D$ is the diffusion time and $s$ describes the aspect ratio of the confocal volume. Plus and minus signs in the second factor in the nominator indicate autocorrelation functions (+) and cross-correlation functions (−). The FRET correlation function of the nsFCS experiment was fitted with the function

$$g_E(\tau) = \left(1 - c_{ab}e^{-t/\tau_{ab}}\right)\left(c_b e^{-t/\tau_b} + c_T e^{-t/\tau_T} + c_S e^{-t/\tau_S}\right) \tag{24}$$

Here, subscripts have the same meaning as in Eq. 23.

The species-specific lifetime-fFCS functions of the protein U2AF2 (Fig. 6e) were computed by laboratory#2 in Agam et al.[56]. We show the fits obtained in this publication. Details about the fitting routine can be found in the supplementary information of ref. 56. The FRET correlation functions for the apo- and holo-state were separately fitted with a sum of three exponential decays. All fits were performed with the nonlinear fitting routine of Mathematica 13.2.

**Microscope and experimental analysis**

**Single-molecule experiments.** The smFRET experiments were performed using a MicroTime 200 (PicoQuant) equipped with an Olympus IX73 inverted microscope. Linearly polarized light from a 485 nm diode laser (LDH-D-C-485, PicoQuant) and unpolarized light at 594 nm from a supercontinuum light source (Solea, PicoQuant) were used to excite donor and acceptor alternately with a repetition rate of 20 MHz in an alternating manner. For nsFCS experiments (Fig. 8), we used a continuous excitation of the donor only. In both cases, the excitation light was guided through a major dichroic mirror (ZT 470-491/594 rpc, Chroma) to a ×60, 1.2 NA water objective (Olympus) that focused the beam into the sample. A home-made sample cuvette was used with a volume of 50 μl made of round quartz cover slips of 25 mm diameter (Esco Optics) and borosilicate glass 6 mm diameter cloning cylinder (Hilgenberg) using Norland 61 optical adhesive (Thorlabs). We performed all measurements (except the nsFCS experiments in Fig. 8) in PIE mode with a laser power of 100 μW (485 nm) and 20 μW (594 nm) measured at the back aperture of the objective. The repetition rate for one period was 20 MHz. Each period consisted of an excitation pulse of the donor laser (485 nm) at the beginning followed by an excitation pulse of the acceptor laser (594 nm) 25 ns after the donor pulse. The photons emitted from the sample passed through the same objective and after passing the major dichroic mirror (ZT 470-491/594 rpc, Chroma), the residual excitation light was filtered by a long-pass filter (BLP01-488R, Semrock). The light was then focused on a 100 μm pinhole. The sample fluorescence was detected with four channels. Donor and acceptor fluorescence was separated via a dichroic mirror (T585 LPXR, Chroma) and each color was focused onto a single-photon avalanche diode (SPAD) (Excelitas) with additional bandpass filters: FF03-525/50, (Semrock) for the donor SPAD and FF02-650/100 (Semrock) for the acceptor SPAD. The arrival time of every detected photon was recorded with a HydraHarp 400 M time-correlated single photon counting module (PicoQuant) at a resolution of 32 ps. All measurements were performed at 23 °C.

SmFRET experiments on the Holliday junction were measured in 1× PBS buffer (8.1 mM sodium phosphate dibasic, 1.5 mM potassium phosphate monobasic, 2.7 mM KCl, 137 mM NaCl, pH 7), 20 mM di-thio-threitol (DTT), 0.001% Tween-20, and varying concentrations of $MgCl_2$. Experiments on DtpA were measured in 20 mM sodium phosphate pH 7.5, 150 mM NaCl, 0.002% LMNG (lauryl-maltose-neopentyl-glycol), and 20 mM DTT. Experiments with $\Delta$Myc were performed in 20 mM TrisHCl pH 8, 2 M GdmCl, 100 mM βmercapto-ethanol, and 0.001% Tween-20. Bursts were identified according to standard routines[43,76] using a photon threshold of 100 (Holliday junction), 50 (DtpA), and 150 ($\Delta$Myc).

**Burst selection and photobleaching filter.** The raw photon numbers for donor $(n'_D)$ and acceptor $(n'_A)$ were corrected for background, quantum yields and detection efficiencies of the microscope $(\gamma = 1.12 \pm 0.09)$, cross-talk $(\beta_{DA} = 0.050 \pm 0.003$ and $\beta_{AD} = 0.0021 \pm 0.0004)$, and acceptor direct excitation $(\alpha = 0.049)$, leading to the corrected photon numbers $n_{DD}$ and $n_{DA}$, where the first subscript indicates the excitation and the second indicates emission. The corrected mean FRET efficiency (temporal average over the burst duration) of a burst is then

$$\left\langle \widetilde{E} \right\rangle = \frac{n_{DA}}{n_{DD} + n_{DA}} \tag{25}$$

For the Holliday junction, DtpA, and the results of the simulations including photobleaching (Fig. 4a–c, right), bursts containing no acceptor fluorescence after acceptor direct excitation (donor-only) were removed by only selecting bursts with a dye stoichiometry ratio

$0.25 \leq S \leq 0.75$ where the stoichiometry of a burst is defined by

$$S = \frac{n_{DD} + n_{DA}}{n_{DD} + n_{DA} + \gamma_{PIE} n_{AA}} \quad (26)$$

Here, $\gamma_{PIE} = 2 - 2.5$ is a factor that accounts for the different excitation intensities for donor and acceptor. The remaining bursts were additionally filtered to exclude bursts in which the acceptor bleached during the transit through the confocal volume. To this end, we only selected those bursts for which the mean arrival times after donor $\langle t_{Dex} \rangle$ and acceptor $\langle t_{Aex} \rangle$ excitation were similar[77]. We then define the burst asymmetry as

$$\alpha_{PIE} = \langle t_{Dex} \rangle - \langle t_{Aex} \rangle \quad (27)$$

We chose a very restrictive threshold of $\alpha_{PIE} \leq 0.05 - 0.1$ ms to exclude photobleaching artifacts in the FRET correlation functions.

**Calculation of FRET correlation functions from photon streams.** FRET correlation functions for experimental and simulated photon data used in this study were calculated with a script written in Mathematica 13.2 (see "Data and Code availability"). After burst selection, all possible photon pairs of each type (AA, AD, DA, DD) were collected from each burst. Those pairs were binned according to their time lag $\tau$. As a result, we obtain the photon pair numbers $N_{XY}(\tau)$ required in Eq. 2. The average apparent FRET value in Eq. 2 was calculated as $\langle E \rangle = n_A / (n_A + n_D)$ where $n_A$ and $n_D$ are the total number of raw acceptor and donor photons in all selected bursts, respectively. We used uniform time lag binning in all cases except the ΔMyc data (Fig. 8), where logarithmic binning was applied. When studying nanosecond dynamics for ΔMyc, a 4-channel setup with two donor and two acceptor detectors was used. To remove nanosecond artefacts from detector deadtimes and afterpulsing, only pairs of photons arriving at different detectors were counted. In all other 4-channel data in which FRET correlation functions were computed down to the shortest lag time of $1 \mu s$, we collect all photon pairs including those from the same detector. A computationally efficient algorithm for calculating the FRET correlation function, using a method previously developed for FCS[78] is available in the latest version of the FRETICA package (see "Code availability").

**Design and labelling of the Holliday junction construct.** Single-stranded DNA (ssDNA) constructs (Supplementary Table 3) were synthesized by IDT (Integrated DNA Technologies, Inc.). Internal amino-modified C6-dT was used to site-specifically label two ssDNA strands by incubating the strands with AlexaFluor488 and AlexaFluor594 NHS-ester (ThermoFisher) for 30 min at room temperature. Labelled strands were separated from unlabelled strands and unreacted dye using reversed-phase chromatography. To this end, a ZORBAX Eclipse Plus C18 ($3.5 \mu m$) column (Agilent) was used, equilibrated with TEAA buffer (0.1 M trimethylamine buffered with acetic acid glacial to pH 6.5 and supplemented with 5% acetonitrile). A gradient of 7–30% acetonitrile over 40 ml was used for the separation. Labelled fractions were dried overnight (SpeedVac) and dissolved in $50 \mu l$ ddH$_2$O. DNA and dye concentrations were quantified by UV-VIS photometry using the extinction coefficients of the free dyes $\varepsilon_{495} = 73,000$ M$^{-1}$ cm$^{-1}$ (AlexaFluor 488) and $\varepsilon_{590} = 92,000$ M$^{-1}$ cm$^{-1}$ (AlexaFluore 594). Only fractions containing 1:1 ratio of dye:ssDNA were used for smFRET experiments. Annealing of ssDNA donor and acceptor strands for smFRET experiments was obtained by mixing the ssDNA strands at a stoichiometric ratio of 0.7:1:1:1 (donor:acceptor:unlabelled:unlabelled) in 50 mM Tris pH 7.5, 0.1 M NaCl, 3 mM Mg$_2$Cl. The samples were heated to 95 °C and cooled gradually over 1 h in a PCR cycler.

**Reporting summary**
Further information on research design is available in the Nature Portfolio Reporting Summary linked to this article.

## Data availability
The experimental data of our work together with a demonstration code to compute FRET correlation functions in the Mathematica Package 13.2 is available under https://doi.org/10.5281/zenodo.15168294. Source data are provided with this paper.

## Code availability
The single-molecule analysis code is accessible as a custom WSTP add-on for Mathematica (Wolfram Research) at https://schuler.bioc.uzh.ch/programs/. A demonstration code is provided under https://doi.org/10.5281/zenodo.15168294.

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

## Acknowledgements

We thank Dima Makarov for critical discussions and helpful comments on an earlier version of this manuscript. This work was supported by a grant of the European Research Council (Grant No. 864578) to H.H.

## Author contributions

I.T. and H.H. designed the research. I.T., D.N., G.R., and H.H. performed research. R.V., I.C., T.L.M., K.B., and C.L. provided experimental data. D.N. provided the implementation in the Fretica package.

## Competing interests
The authors declare no competing interests.
