## [Transparent Peer Review file · Nature Communications]

Model-free photon analysis of diffusion-based single-molecule FRET experiments

Corresponding Author: Professor Hagen Hofmann

Version 0:

Reviewer comments:

Reviewer #1

(Remarks to the Author)

The paper presents a novel, model-free method to analyze photon trajectories in diffusion-based single-molecule Förster Resonance Energy Transfer (smFRET) experiments. Traditional analysis often relies on Markov models to describe protein dynamics, which can introduce biases due to the projection of complex conformational spaces onto a single FRET efficiency coordinate. The authors introduce an alternative that computes FRET efficiency correlation functions directly from photon data, mitigating artifacts caused by diffusion and the finite photon trajectory length. The manuscript studies this concept with numerical simulations and several experimental applications that validate this method across micro- to millisecond timescales, demonstrating its broad applicability to biomolecular dynamics.

Specifically, the authors address a significant limitation in conventional smFRET by developing a model-independent technique that extracts dynamic information without imposing Markovian constraints. The manuscript lays out the mathematical foundations of FRET correlation functions and use a combination of simulated Brownian dynamics and real data (from membrane proteins, DNA, and disordered proteins) to demonstrate the efficacy of the method. In particular, the use of correlation ratios allows for detecting photobleaching artifacts and to identify triplet dynamics which is usually difficult to achieve in model-based Markov analysis. One of the significant advantages of the propose new data analysis is its broad applicability because it does not require any a priori knowledge of the dynamics underlying the studied biomolecular system, as nicely demonstrated in its application to both membrane proteins and intrinsically disordered proteins (IDPs). I strongly recommend publication of the manuscript after some minor revisions.

In particular:

1. On line 119 on page 4: Should it not read "lag times from τ to $\tau + \Delta\tau$ " instead of "lag times from τ to $\Delta\tau$ "?
2. At line 151 on page 5, the authors write: "The simulations assumed Poisson photon emission statistics that is correct at sufficiently low excitation rates and at timescales slower than the fluorescence lifetimes of the dyes." Probably my question is trivial, but why should the Poisson photon statistics break down at high count rates? Due to antibunching?

Reviewer #2

(Remarks to the Author)

The work of Terterov et al. presents a new model-free analysis method dedicated to the extraction of kinetic information available from single molecule Förster resonance energy transfer (FRET) experiments. Based on simulated data the authors introduce their approach for calculating FRET correlation functions and correlation ratios, which allows for accurate identification of dynamic time scales up to several milliseconds. This type of analysis and especially the use of FRET correlation functions can reveal discrepancies when imposing Markov models (e.g. by comparing the non-exponential correlation function to the single exponential derived from a 2-state Markov model). Thus this methodology can be used to validate model based approaches. Importantly, based on the simulated single molecule data (by means of correlation ratios) the authors also explore the effects of experimental artifacts, such as photobleaching and differences in dye brightness, on the analysis. The authors then apply their analysis method on 3 suitable experimental systems, namely a membrane protein DtpA, various lengths DNA duplexes for which a millisecond dynamics timescale is extracted and an intrinsically disordered protein variant of Myc, for which sub-ms unfolded chain dynamics is obtained without the interference of diffusion-related components.

This new analysis method developed by Terterov et al. has a key advantage over existing model-free analysis methods as it requires only a few thousand molecules, reducing measurement time, in striking contrast to other methods for example the recurrence analysis (RASP) which often requires >10 hours of acquisition time. Importantly also this method allows identification of static heterogeneity present in the sample, photobleaching artifacts, and also validation of model-based fitting approaches. I believe that this analysis method will be used by many groups specializing in the field of diffusion-based single-molecule FRET experiments. The manuscript is well written and provides the necessary code for implementing this analysis on other systems by other groups.

I thus recommend this work for publication in Nature Communications as it is.

Reviewer #3

(Remarks to the Author)

I am a theorist, and I have studied and analyzed single-photon and binned smFRET experiments using hidden Markov Models (HMMs) many times in the past. With my background, I was able to mostly understand the paper. The contribution to literature is significant and worth publication in Nat. Comm. However, I have minor comments regarding presentation and pedagogy.

1. It may be helpful to have subscripts or different symbols for true continuous FRET efficiency and corrected FRET efficiency....The two epsilons are visually similar and can lead to confusion.
2. Fig 1. Diffusion gray color is not immediately perceptible in panel d. Some other color choice or may be slightly darker colors may help
3. Fig 2. Typo....."indicted" vs "indicated"
4. Fig 2. The white lines are confusing, especially with the white background of the page. It took me some time to understand what authors are talking about.
5. Fig 2. Why not keep the number of bursts fixed in panel a?
6. Fig 2, panel d. Not immediately clear which color corresponds to which concentration. Legend colors are too similar.
7. Fig 2: it would be nice to know the average time spent in the confocal volume in caption or the diffusion coefficient used for the simulation.
8. Line 163. For pedagogical reasons, it would be nice to explain why the apparent relaxation rate would be $\lambda = 2k$.
9. Line 136. Do the authors mean..."to large extent"?
10. Line 171. It would be helpful to assign a symbol to common timescales of physical significance instead of saying "slower than milliseconds". It puts the timescales in physical context. The "milliseconds" may be obvious to experts but for a wider audience, physical meaning is more easy to understand. This would be helpful everywhere in the paper and help with the generality of the method. (edited)
11. Line 179. Please elaborate more on advantages of performing experiments at high concentrations. The explanation doesn't have enough physical reasoning for a non-expert reader.
12. Fig 3, panel a. Not sure what gray shaded area authors are talking about. Visually, it is not very distinct.
13. Lines 230 & 241. Authors mention 2 state model provides excellent fit and mention recoloring. I am not sure what recoloring means. I am not sure how the 2 state model produces good fit in Fig 3c. Actually, it would be helpful to define what fitting means here. I have worked with a lot of binned smFRET data. In the binned case, typically, when there is state degeneracy as in the 4-state model here, the histograms are independent of degeneracy and do not resolve the non-Markovianity as the authors note here as well.

I am guessing authors are trying to do something similar here but non-Markovianity results in different histograms (panels a and c). So how they manage to get same fits.
I am sure what authors doing here is correct but the writing needs to improve for pedagogy.
14. It may be useful the analysis starting line 257 for photobleaching using the degenerate model. In panel c, bleaching results in correlations that have similar behavior as in the degenerate model (naively) in Fig 3 (the correlation data lies above the theoretical black line). How do we distinguish degeneracy from photobleaching then? I guess other cross-correlation curves with asymmetry help?

Reviewer #4

(Remarks to the Author)

The authors developed a photon-by-photon correlation approach to extract dynamic information from fluctuations in FRET efficiency. The authors demonstrate the principle of the technique and investigate the influences of various parameters on the method. Afterwards, they demonstrate the capabilities of the technique on experimental systems. The approach is exciting and powerful. The overall flow of the paper is well organized. However, some details are missing at various places. The description of the methods and some graphs are difficult to follow, especially for non-experts. Also, the presentation of the experimental data should be reconsidered to improve the didactic presentation of the method. Upon addressing these issues, the paper can be considered for publication.

Major points

- 1) The figures and details are often difficult to follow for non-specialists. While this may be acceptable for a more specific journal, for publication in Nature Communications the article should be more accessible to a general audience.
- 2) I would recommend moving the derivation of the FRET correlation curve to the methods section of the main text, as this is the heart of the paper.
- 3) The work would benefit from a comparison to the rates determined using the FRET correlation approach to that of other approaches such as species FCCS, dynamic pda or sliding-gate FCS.
- 4) Experimental details are sometimes missing. Sample, measurement times, PIE/no PIE...
- 5) The Supplementary Information could use more structure to make it easier to follow the vast amount of information. For example, by adding Supplementary Notes and Sections.
- 6) E.g. Line 69: The authors like to stress the analysis as a model-free approach. I appreciate what the authors wish to say but, in the end, they use models (e.g. in Figure 6d) to evaluate the kinetics and extract the rates. Hence, I would suggest the authors to be more accurate and cautious with the wording.
- 7) Eq 1: Although the authors have made efforts to make the concept of the FRET correlation function understandable, more efforts are needed. If I follow correctly, $N(\tau)$ is the non-normalized correlation function of all photons, which should approach the average photon rate (random possibility of detecting photons), which should not zero. (in this sense, the inset in panel d is deceiving). The values depend on background, molecule concentration, brightness, etc (information missing in the figure caption of Fig 1, see below).
- 8) Figure 1: More details are needed in Figure 1. What is the sample (or is it simulated data), what are the count rates, concentration etc. Also, panel b is not very clear about what the authors wish to communicate with this panel and more clarification would be useful.
- 9) Figure 2: Adding simulations with non-symmetric rates to compare would be useful. This can be added to the SI.
- 10) Figure 4f,g; Lines 313-314: 'Contrary to the relaxation time, the amplitude of $g_{\epsilon}(x)$ is strongly affect...' The deviations in Figure 4f and 4g look similar. Hence, this statement seems unfounded. Please elaborate.
- 11) Figure 6: As a proof of principle, the data in Figure 6 are much too complicated. The authors should consider first presenting a more simple system (e.g. a DNA hairpin or something similar) for the first experimental data, and then follow up with the data in Figure 6.
- 12) Figure 7: My recommendations to the authors would be to remove this figure. It is too complicated and does not help the overall paper. Also, I do not agree from the fit residuals that three exponentials are warranted. If the authors wish to keep the figure, then they should a) explain why the correlation curves misbehave for the 6 bp curves (panel b), give the rates somewhere and c) highlight other distances that then quasi no FRET (39 bp) and Dexter/quenching (2 bp) samples. That the 2 bp separation sample behaves differently is not surprising. d) The spectrum of Alex488 changes with pH, how does this influence their results?

Minor points

- Line 39-40: "At the level of individual molecules, motions are stochastic and driven by thermal noise." This is often the case, but not always (e.g. molecular motors). Please change to "motions are often stochastic..."
- Lines 104-110: "We therefore distinguish three FRET efficiencies: the apparent FRET efficiency E , defined by the raw photon counts of donor and acceptor, the FRET efficiency ϵ , computed from the photon counts corrected for background, relative dye brightness and instrumental imperfections, and the continuous true FRET efficiency ϵ_s , which is given by the time-dependent donor-acceptor distance $r(t)$ and the dye-specific Förster-distance R " (Supplementary Information eq. S1)." The definition of a continuous true FRET efficiency is confusing. What I believe the authors are trying to say is that, for dynamic systems, the FRET efficiency then fluctuates between states with time. However, this could theoretically also be detectable). In addition, if the orientation of the dyes (or the spectra) are changing, so is the R_0 and then R_0 should be time dependent. The authors need to be more explicit about the difference between ϵ (Mathematical epsilon) and ϵ (Greek epsilon).
- Figure 1a: In English, one reads from left to write. Hence, I suggest flipping the pathway of the molecule through the volume to follow accordingly. (Also in Figure 5c).
- Figure 2a: add 'n=' to figure legends.
- Figure 2d: Please write $2k$ out as $k_{12} + k_{21}$ to be less confusing.
- Lines 173-174: "In fact, static heterogeneity is difficult to spot otherwise and is rarely included in model-based analysis approaches." - E-tau plots are useful for detecting static and dynamic heterogeneities. Please rewrite.
- Figure 4d-j. A discussion of more realistic parameters and mentioning the various values for typical dye pairs would help improve this figure.
- Figure 4d, 4h. What is the meaning of the grey boxes? I find it confusing that the box narrows when the brightnesses don't change with protein conformation, and retains the same width when the brightnesses do change with conformation.
- Figure 5a: For the transition from S0S1 to S1S1. The up arrow should be dark blue.
- Figure 5a, caption: "The first letter refers to the donor and the second letter indicates the state of the acceptor." This is redundant with respect to a previous sentence: "Each state is denoted by the electronic states of donor (first symbol) and acceptor (second symbol)"
- Figure 5d: Which FRET efficiency (i.e. which epsilon) is being plotted here. Not always easy to follow in the text.

Figure 5e, inset: The abscissa axes should be labeled 'Lag Time'.

Figure 5e, caption: This looks very much like nsFCS, but a couple orders of magnitude slower. It would be useful to explicitly highlight the difference in timescale here.

The authors talk about "bleaching" throughout the paper. This should be referred to as "photobleaching".

Figure 6b: Can the triplet state be extracted from these plots and then used to fix the timescale of the triplet kinetics in panel c?

Figure 6c. Please plot both green components and provide the uncertainty in the rates.

Figure 6c,d: Please include details of how the fitting was performed. The 1000 label is shifted in the abscissa axis.

Lines 487-491: "We collected > 800,000 bursts to also probe the nanosecond dynamics sufficient photon pair statistics (Fig. 8a, inset)." The (Fig 8a, inset) is confusing. The nanosecond dynamics are not in the inset, only $n = 847,012$. Hence, the reference is not clear.

Line 575: Please give the diffusion coefficient in $\mu\text{m}^2/\text{s}$.

Lines 619-620: Please give background rates in Hz (or kHz).

Line 624: Here, the authors introduce crosstalk, but do not refer to it in the development of the theory. Please discuss the influence of crosstalk when correcting the correlation function for system imperfections.

Line 669: "Here, n is the total photon emission rate of donor and acceptor." This is unclear. Is this the sum of the donor and acceptor emission rates, or will there be subscripts to denote donor and acceptor? Please clarify.

Lines 675-677: "To minimize the noise in the recolored FRET correlation functions, we recolored the data ten times and computed the average FRET correlation function of the ten recolored data sets." I do not follow, how does this minimize the noise? If this is the same data, how does recoloring it ten times help?

Lines 691-692: "To access the effect of bleaching (Fig. 4a-c), we modelled a system that switches between 4 states:" I assume that switching from the photobleached states back to FRET states is not allowed. Please reword.

Line 710: " $\gamma_{\text{PIE}} = 2$ " Please define what this is.

Line 724: Typo: form should be from.

Line 881: Equation 27. The authors should cite the work where this was introduced over 10 years ago.

Line 987: Typo in reference: Dueller should read Mueller. (Also SI Reference 3).

SI: Throughout the SI, please add subscripts to the $\langle \rangle$ to clarify what dimension is being averaged over.

SI Line 220-222: Eqn S26: Please add subscript D to sigma.

SI Line 366: Typo. Should read "Fig. 1..."

SI Line 444: Typo: "Indepent" -> Independent

Version 1:

Reviewer comments:

Reviewer #1

(Remarks to the Author)

The authors have satisfactorily answered my questions. I recommend now publication as is.

(Remarks on code availability)

Reviewer #3

(Remarks to the Author)

The authors have addressed all my comments on the original paper satisfactorily, improving the pedagogy. I recommend publication.

(Remarks on code availability)

Reviewer #4

(Remarks to the Author)

The authors have admirably addressed the concerns raised by the reviewers. I would suggest flipping Figure 5c to go from left to right as was done in Figure 1. I highly recommend the paper for publication.

(Remarks on code availability)

Reviewer #1 (Remarks to the Author):

The paper presents a novel, model-free method to analyze photon trajectories in diffusion-based single-molecule Förster Resonance Energy Transfer (smFRET) experiments. Traditional analysis often relies on Markov models to describe protein dynamics, which can introduce biases due to the projection of complex conformational spaces onto a single FRET efficiency coordinate. The authors introduce an alternative that computes FRET efficiency correlation functions directly from photon data, mitigating artifacts caused by diffusion and the finite photon trajectory length. The manuscript studies this concept with numerical simulations and several experimental applications that validate this method across micro- to millisecond timescales, demonstrating its broad applicability to biomolecular dynamics. Specifically, the authors address a significant limitation in conventional smFRET by developing a model-independent technique that extracts dynamic information without imposing Markovian constraints. The manuscript lays out the mathematical foundations of FRET correlation functions and use a combination of simulated Brownian dynamics and real data (from membrane proteins, DNA, and disordered proteins) to demonstrate the efficacy of the method. In particular, the use of correlation ratios allows for detecting photobleaching artifacts and to identify triplet dynamics which is usually difficult to achieve in model-based Markov analysis. One of the significant advantages of the propose new data analysis is its broad applicability because it does not require any a priori knowledge of the dynamics underlying the studied biomolecular system, as nicely demonstrated in its application to both membrane proteins and intrinsically disordered proteins (IDPs). I strongly recommend publication of the manuscript after some minor revisions.

We thank the referee for her/his positive opinion on our work.

In particular:

1. On line 119 on page 4: Should it not read “lag times from τ to $\tau + \Delta\tau$ ” instead of “lag times from τ to $\Delta\tau$ ”?

We thank the referee for spotting this mistake. The referee is absolutely correct. We changed the statement in the revised version of the manuscript.

2. At line 151 on page 5, the authors write: “The simulations assumed Poisson photon emission statistics that is correct at sufficiently low excitation rates and at timescales slower than the fluorescence lifetimes of the dyes.” Probably my question is trivial, but why should the Poisson photon statistics break down at high count rates? Due to antibunching?

The reviewer makes an interesting point. Indeed, there is no reason why Poisson statistics for photon emission should break down at high count rates. However, in line 151 we were referring to high excitation rates, not count rates. For organic fluorophores, high excitation rates will lead to more significant population of long-lived triplet states, the presence of which will render the photon emission statistics non-Poissonian, since there will be microsecond long periods without photon emission once the fluorophore is in the triplet state. In addition, the referee is correct in that on short timescales of a few nanoseconds, the Poisson limit is also broken due to antibunching (even at low excitation rates). Although high count rates are generally associated with high excitation rates, it is important to distinguish between the two quantities.

Reviewer #2 (Remarks to the Author):

The work of Terterov et al. presents a new model-free analysis method dedicated to the extraction of kinetic information available from single molecule Förster resonance energy transfer (FRET) experiments. Based on simulated data the authors introduce their approach for calculating FRET correlation functions and correlation ratios, which allows for accurate identification of dynamic time scales up to several milliseconds. This type of analysis and especially the use of FRET correlation functions can reveal discrepancies when imposing Markov models (e.g. by comparing the non-exponential correlation function to the single exponential derived from a 2-state Markov model). Thus this methodology can be used to validate model based approaches. Importantly, based on the simulated single molecule data (by means of correlation ratios) the authors also explore the effects of experimental artifacts, such as photobleaching and differences in dye brightness, on the analysis. The authors then apply their analysis method on 3 suitable experimental systems, namely a membrane protein DtpA, various lengths DNA duplexes for which a millisecond dynamics timescale is extracted and an intrinsically disordered protein variant of Myc, for which sub-ms unfolded chain dynamics is obtained without the interference of diffusion-related components.

This new analysis method developed by Terterov et al. has a key advantage over existing model-free analysis methods as it requires only a few thousand molecules, reducing measurement time, in striking contrast to other methods for example the recurrence analysis (RASP) which often requires >10 hours of acquisition time. Importantly also this method allows identification of static heterogeneity present in the sample, photobleaching artifacts, and also validation of model-based fitting approaches. I believe that this analysis method will be used by many groups specializing in the field of diffusion-based single-molecule FRET experiments. The manuscript is well written and provides the necessary code for implementing this analysis on other systems by other groups.

I thus recommend this work for publication in Nature Communications as it is.

We thank the referee for her/his extremely positive opinion on our work.

Reviewer #3 (Remarks to the Author):

I am a theorist, and I have studied and analyzed single-photon and binned smFRET experiments using hidden Markov Models (HMMs) many times in the past. With my background, I was able to mostly understand the paper. The contribution to literature is significant and worth publication in Nat. Comm. However, I have minor comments regarding presentation and pedagogy.

1. It may be helpful to have subscripts or different symbols for true continuous FRET efficiency and corrected FRET efficiency....The two epsilons are visually similar and can lead to confusion.

We thank the referee for her/his suggestion. Following the advice, we now refer to the corrected FRET efficiency, indicated before by ϵ , as $\tilde{\epsilon}$. This notation also clarifies that the corrected FRET efficiencies are computed from the detected photon stream.

2. Fig 1. Diffusion gray color is not immediately perceptible in panel d. Some other color choice or may be slightly darker colors may help

We darkened the color of the region that indicates the diffusion timescales in Fig. 1d and in the inset and hope that the contrast will be better visible.

3. Fig 2. Typo....."indicted" vs "indicated"

We corrected the typo.

4. Fig 2. The white lines are confusing, especially with the white background of the page. It took me some time to understand what authors are talking about.

We changed the color of the data and used black lines to indicate the fits.

5. Fig 2. Why not keep the number of bursts fixed in panel a?

We thank the referee for bringing this up. Of course, we could keep the number of bursts fixed, and we discussed this internally before the initial submission. However, our intention was to be as realistic as possible in our simulations. Therefore, we ran the simulations at different exchange rates for the same time (500 s) and at the same particle concentration as we would do in an actual experiment. Since diffusion is a stochastic process, this leads to slightly different numbers of identified bursts. However, these variations are extremely small (relative: 1%) and do not affect the quality (noise) of the FRET correlation functions shown in Fig. 2d.

6. Fig 2, panel d. Not immediately clear which color corresponds to which concentration. Legend colors are too similar.

We thank the referee for pointing out this difficulty. We now changed the colors such that the identification of points with concentrations should be clear now.

7. Fig 2: it would be nice to know the average time spent in the confocal volume in caption or the diffusion coefficient used for the simulation.

We now provide the average burst duration of the simulations in the caption of Fig. 2.

8. Line 163. *For pedagogical reasons, it would be nice to explain why the apparent relaxation rate would be $\lambda = 2k$.*

We agree with the referee and now provide a short explanation of the relationship between apparent relaxation rate and the Eigenvalues of the kinetic rate matrix.

9. Line 136. *Do the authors mean... "to large extent"?*

Indeed, the referee is absolutely correct. We apologise for the misspelling and now changed the wording.

10. Line 171. *It would be helpful to assign a symbol to common timescales of physical significance instead of saying "slower than milliseconds". It puts the timescales in physical context. The "milliseconds" may be obvious to experts but for a wider audience, physical meaning is more easy to understand. This would be helpful everywhere in the paper and help with the generality of the method. (edited)*

We agree with the reviewer that a single symbol indicating the relevant timescale might facilitate understanding. Although several timescales are involved in the emission of photons (singlet and triplet lifetimes of the dyes), the most relevant timescale for our purpose is the timescale of diffusion of molecules through the confocal volume. We will now denote this timescale by the symbol t_D and make it clear that this timescale is on the order of 1 ms.

11. Line 179. *Please elaborate more on advantages of performing experiments at high concentrations. The explanation doesn't have enough physical reasoning for a non-expert reader.* 12. Fig 3, panel a. *Not sure what gray shaded area authors are talking about. Visually, it is not very distinct.*

In fact, there are not many advantages to performing single molecule experiments at higher concentrations. The phrase "in some cases it is advantageous to perform smFRET experiments at higher concentrations" may have caught the reviewers' attention. Upon reflection, we now believe that it would be more appropriate to say that in some cases it is unavoidable (rather than advantageous) to work at higher concentrations. We now clarify the two scenarios better: (i) nsFCS - here higher concentrations lead to more photons and thus to better signal-to-noise in the nsFCS curves and (ii) binding assays measured by intermolecular FRET - here both binding partners are labeled and slightly higher concentrations than usual in smFRET experiments (50pM is the standard) might be needed to increase the amount of complex between the two partners. In our lab, we have never seen another scenario where higher concentrations would be needed.

We acknowledge that the gray shaded area (it is actually a gray band) in Fig. 3 a,c is difficult to see. This area represents 1SD of the recoloring. We have now darkened the area and added a black frame around it to make it more visible.

13. Lines 230 & 241. *Authors mention 2 state model provides excellent fit and mention recoloring. I am not sure what recoloring means. I am not sure how the 2 state model produces good fit in Fig 3c. Actually, it would be helpful to define what fitting means here. I have worked with a lot of binned smFRET data. In the binned case, typically, when there is state degeneracy as in the 4-state model here, the histograms are independent of degeneracy and do not resolve the non-Markovianity as the authors note here as well.*

I am not sure what recoloring means.

The recoloring procedure for diffusive smFRET data was introduced by Irina Gopich and Attila Szabo¹. The idea is to simulate for an existing photon stream (measured or simulated) new ‘colors’ (donor/green, acceptor/red) for the photons while keeping their detection times. We explained the procedure on p. 21 of the Methods section and have now made our explanation more explicit (p. 23 revised manuscript). The recoloring is based on the assumed kinetic model of interconverting states. Using the rate coefficients obtained from the likelihood maximization, we simulate a time-trajectory of conformational states for the given sequence of photon detection times of a burst. For example, the model in Fig. 3a would be a two-state model and the stochastic trajectory would show switching between these two states. In a second step, new photon colors are chosen randomly according to the apparent FRET efficiency attributed to the current conformations.

Out of the resulting re-colored bursts, we generate a new transfer efficiency histogram. Typically, we repeat this ten times and then plot the average histogram as a black line in the figure.

Actually, it would be helpful to define what fitting means here.

We described the fitting procedure on p.20 (Non-Markov model and likelihood maximization) in the original manuscript. In short, using a model, a 2-state model for the data in Fig. 3, we numerically maximized the log-likelihood function (eq. 14) calculated for the photon streams from all bursts in a simulation with respect to the model parameters (the rate k , the FRET efficiencies ε_1 and ε_2 , and the total photon rate of donor and acceptor n).

I am guessing authors are trying to do something similar here but non-Markovianity results in different histograms (panels a and c). So how they manage to get same fits. I am sure what authors doing here is correct but the writing needs to improve for pedagogy.

Clearly the fits to the two simulations are different, but the fit quality (BIC), a parameter that is often used to judge whether a model is fitting the data, is identical between the two data sets. Our statement was that it is not always possible to identify a wrong model using the quality of the fit.

In addition, non-Markovianity leads to different histograms in Fig. 3a,c because we use the same value of the rate constant k (5 ms^{-1}) in the simulation of the 2-state and 4-state model. Since the non-Markov model has four states, two states with low and two states with high FRET efficiency, the time spent at high or low FRET efficiency is longer than in the 2-state model with the same interconversion rate k . In fact, one can show that the apparent rate constant of switching between low and high FRET efficiency would be $k/2$ in the 4-state non-Markov model: Using the nomenclature in Fig. 3c (four states denoted as $1'$, 1 , 2 , $2'$), the flux from state 1 to state 2 is $J = kp_1$ where $p_1 = 1/4$. Now, assuming an apparent 2-state model of switching between an apparent low FRET and a high FRET state with identical forward and backward rate constants k_{app} , the flux from low to the high FRET state would be $J = k_{app}p_{lowFRET}$ with $p_{lowFRET} = 1/2$. Equating the two fluxes, we find $k_{app} = k/2$. Hence, a fit of the 4-state non-Markov model with an apparent 2-state model should provide the apparent rate $k_{app} = 2.5 \text{ ms}^{-1}$, which is close to the value found (2.1 ms^{-1}). The reason why the fitted value is smaller than the expected value is due to the limited photon emission rate, a problem of any Hidden-Markov analysis that has been discussed extensively in ref². This lower apparent switching rate between low- and high-FRET states explains the broader FRET histogram in Fig. 3c compared to Fig. 3a. We now provided this explanation in the main text of the manuscript.

We now provide a few more sentences to explain the difference in the FRET histograms between Markov- and non-Markov case on p.7-8 of the revised manuscript.

14. It may be useful the analysis starting line 257 for photobleaching using the degenerate model. In panel c, bleaching results in correlations that have similar behavior as in the degenerate model (naively) in Fig 3 (the correlation data lies above the theoretical black line). How do we distinguish degeneracy from photobleaching then? I guess other cross-correlation curves with asymmetry help?

We thank the reviewer for this comment. She/he is absolutely correct. The way to differentiate between a complicated kinetic model and photobleaching would be to analyse the symmetry in the cross-correlation ratios. If they are asymmetric, it cannot be concluded that a slow decay in the FRET correlation function is the result of slow conformational dynamics. We now provide an additional statement at the end of the photo-bleaching paragraph that uses this point as motivation for checking the cross-correlation ratios (p. 9).

Reviewer #4 (Remarks to the Author):

The authors developed a photon-by-photon correlation approach to extract dynamic information from fluctuations in FRET efficiency. The authors demonstrate the principle of the technique and investigate the influences of various parameters on the method. Afterwards, they demonstrate the capabilities of the technique on experimental systems. The approach is exciting and powerful. The overall flow of the paper is well organized. However, some details are missing at various places. The description of the methods and some graphs are difficult to follow, especially for non-experts. Also, the presentation of the experimental data should be reconsidered to improve the didactic presentation of the method. Upon addressing these issues, the paper can be considered for publication.

We thank the referee for her/his positive opinion on our work. Below, we address the points raised by the referee.

Major points

1) The figures and details are often difficult to follow for non-specialists. While this may be acceptable for a more specific journal, for publication in Nature Communications the article should be more accessible to a general audience.

Based on the reviewers detailed comments below and also following the suggestion of reviewer 3, we changed the outlook of many figures to help distinguish colors better. In Fig. 1, we added boxes with headers and additional information to increase understanding of the individual steps of the analysis.

2) I would recommend moving the derivation of the FRET correlation curve to the methods section of the main text, as this is the heart of the paper.

We thank the reviewer for this kind comment. We have tried to do this, but given the size limit of the methods section given by the journal, we will not be able to fit the derivation in there. We therefore hope for the referee's understanding. We believe that demonstrating the principle of FRET correlation together with its application to simulated and experimental data, is the most important step to foster understanding and to motivate use of the new tool.

3) The work would benefit from a comparison to the rates determined using the FRET correlation approach to that of other approaches such as species FCCS, dynamic pda or sliding-gate FCS.

We thank the reviewer for this helpful comment. We now used published data from a benchmark study (Agam et al. ref. ³) where data have been analysed using FCCS. We downloaded the raw data and re-analysed them using our approach of FRET correlation functions. Overall, our results confirm the dynamics identified in this benchmark study not only qualitatively but also quantitatively. These results are now shown in the new Fig. 6e-g or the revised manuscript.

4) Experimental details are sometimes missing. Sample, measurement times, PIE/no PIE...

We apologise for the missing information. We provided the experimental information in the methods section but realized that indeed information was missing. We added this information now in the Methods under *Microscope and experimental analysis*.

5) *The Supplementary Information could use more structure to make it easier to follow the vast amount of information. For example, by adding Supplementary Notes and Sections.*

We thank the reviewer for this comment. We now clarified the SI structure better by dividing it into five supplementary notes. Since the guidelines of the journal pose some restrictions (we were indeed pointed at this fact by the editor), we are afraid that we cannot have sections. However, we now provide a table of content on the first page of the SI, including the titles of the individual chapters and we highlight better the beginning of a new chapter. We hope that at least the current structure of the SI is now better understandable.

6) *E.g. Line 69: The authors like to stress the analysis as a model-free approach. I appreciate what the authors wish to say but, in the end, they use models (e.g. in Figure 6d) to evaluate the kinetics and extract the rates. Hence, I would suggest the authors to be more accurate and cautious with the wording.*

We thank the reviewer for this comment. What we meant to say is that we do not need a model beforehand to compute FRET correlation functions. In fact, as with any other correlation function, one can almost read the timescale by eye, just by looking at the function. Of course, precise numbers require fitting and we agree with the referee in that even an exponential fit is essentially a model. Yet, an empirical multi-exponential fit represents a whole class of models and differentiating between them will be virtually impossible based on the FRET correlation function alone. We have now clarified what we actually mean with the term model-free in the introduction of the revised manuscript.

7) *Eq 1: Although the authors have made efforts to make the concept of the FRET correlation function understandable, more efforts are needed. If I follow correctly, $N(\tau)$ is the non-normalized correlation function of all photons, which should approach the average photon rate (random possibility of detecting photons), which should not zero. (in this sense, the inset in panel d is deceiving). The values depend on background, molecule concentration, brightness, etc (information missing in the figure caption of Fig 1, see below).*

We thank the reviewer for this comment. We now clarified the relationship between the photon pair number distributions ($N_{XY}(\tau)$ and $N(\tau)$) and intensity correlations better in the revised manuscript. First, the equal sign in eq. 3 was now removed and we replaced it by an approximately equal sign. Importantly, however, eq. 3 only states that the ratio of $N_{XY}(\tau)$ and $N(\tau)$, namely $N_{XY}(\tau)/N(\tau)$, is approximately equal to the ratio of the intensity correlation function. Since $N_{XY}(\tau)$ is computed from identified bursts, i.e., short photon streams, it will not be identical to the intensity correlation function that one would compute for the whole measurement including the periods of pure background (without a burst). However, for a single burst, $N(\tau)$ is indeed related to the intensity correlation function. Briefly, the intensity correlation function of a signal I is given by $g_I(\tau) = \frac{1}{T-\tau} \sum_{t=0}^T I(t)I(t+\tau)$ whereas the pair number is $N(\tau) = \sum_{t=0}^T I(t)I(t+\tau)$. Hence, the difference between both quantities is the normalization factor $\frac{1}{T-\tau}$. For very long trajectories, both $g_I(\tau)$ and $N(\tau)$ should not decay to zero as correctly noted by the referee. However, given that the identified bursts have a finite length, the pair number distributions $N_{XY}(\tau)$ and $N(\tau)$ will unavoidably drop to zero for lag times much longer than the burst duration.

8) *Figure 1: More details are needed in Figure 1. What is the sample (or is it simulated data), what are*

the count rates, concentration etc. Also, panel b is not very clear about what the authors wish to communicate with this panel and more clarification would be useful.

We thank the reviewer for this comment and re-designed Fig. 1 to obtain a better graphical representation of how the FRET correlation is being calculated. Importantly, all panels in figure 1 are schematic to highlight the different steps in the analysis together with the features of the photon pair distributions (panel c) and of the FRET correlation function (panel d). Panel b only illustrates the individual pairs of photons that can be extracted from the photon streams of each burst. We added explanations in the figure caption to clarify the procedure.

9) Figure 2: Adding simulations with non-symmetric rates to compare would be useful. This can be added to the SI.

We thank the reviewer for this comment and added simulations of the two-state model with different forward and backward rate constants in the Supplementary Figure 5.

10) Figure 4f,g; Lines 313-314: 'Contrary to the relaxation time, the amplitude of $g_e(\tau)$ is strongly affect...' The deviations in Figure 4f and 4g look similar. Hence, this statement seems unfounded. Please elaborate.

The difference between Fig. 4f and 4g is the y-axis, which is the relative deviation in % from the true rate constant in 4f and the normalized amplitude (normalized by the true amplitude). A value of 5% in 4f indicates that the rate constant is only 5% different from the true rate constant. A value of 0.1 in 4g indicates that the amplitude is only 10% of the true amplitude. At the extreme γ -values (0.1 and 10), the rate constant only differs by 6% from the true rate constant whereas the amplitude is only 5% of the true value, i.e., it differs by 95%.

11) Figure 6: As a proof of principle, the data in Figure 6 are much too complicated. The authors should consider first presenting a more simple system (e.g. a DNA hairpin or something similar) for the first experimental data, and then follow up with the data in Figure 6.

We thank the referee for this comment and followed his advice. We labeled and purified dsDNA for a Holliday junction (HJ) that has previously been used to demonstrate the applicability of H²MM in extracting dynamics from experiments⁴. HJ switches between equally populated high- and low-FRET states with rates that depend on the concentration of MgCl₂ in solutions. We performed PIE-experiments at different concentrations of MgCl₂ and computed the FRET correlation functions. As expected, the FRET correlation functions show two decays. The fast decay at a timescale of 15 μ s is due to the known triplet dynamics of the dyes and fast conformational fluctuations of the DNA. The slow decay is due to the switching between the two states. The apparent rate constants of the slow decay agree well with the rates obtained from the analysis with H²MM. We now discuss these data in the revised Fig. 6 as an introduction into the use of FRET correlation functions in the analysis of real experimental data.

12) Figure 7: My recommendations to the authors would be to remove this figure. It is too complicated and does not help the overall paper. Also, I do not agree from the fit residuals that three exponentials are warranted. If the authors wish to keep the figure, then they should a) explain why the correlation curves misbehave for the 6 bp curves (panel b), give the rates somewhere and c) highlight other distances that

then quasi no FRET (39 bp) and Dexter/quenching (2 bp) samples. That the 2 bp separation sample behaves differently is not surprising. d) The spectrum of Alex488 changes with pH, how does this influence their results?

We thank the referee for this comment. Since we now added the experiments and analysis of the Holliday junction and furthermore added an analysis to compare FRET correlation functions to FCCS, we agree with the referee and removed the dsDNA data from the manuscript.

Minor points

Line 39-40: "At the level of individual molecules, motions are stochastic and driven by thermal noise." This is often the case, but not always (e.g. molecular motors). Please change to "motions are often stochastic..."

At this point, we disagree with the referee. In fact, we would claim that the term ‘molecular motors’ is misleading. In fact, every process at nanometer lengths scales is driven by thermal noise. For instance, also the motions of molecular motors are stochastic. In our view, the alternative to stochastic motions are deterministic motions. However, even for highly processive walkers (kinesin), the dynamics are stochastic despite the fact that the walker moves in one direction. It cannot deterministically be predicted when exactly the walker will make the next step. Hence, a reaction can be directional (break detailed balance) under non-equilibrium conditions (enzymes etc.), yet the dynamics of a single molecule will be stochastic. In fact, the question of whether and how non-equilibrium conditions can be inferred from the stochastic molecular motions of an enzyme is a topic of intense discussion. For example, Hugel and co-workers showed experimentally that the chaperone Hsp90 breaks detailed balance only slightly while it is turning over ATP⁵. We also would provide some of the more recent theoretical discussion about this topic with reference⁶. To summarize, albeit non-equilibrium conditions can break detailed balance in the kinetic scheme of conformational changes in enzymes and motors, the driving force for switching between different conformations remains the thermal noise.

Lines 104-110: "We therefore distinguish three FRET efficiencies: the apparent FRET efficiency E , defined by the raw photon counts of donor and acceptor, the FRET efficiency ϵ , computed from the photon counts corrected for background, relative dye brightness and instrumental imperfections, and the continuous true FRET efficiency ε , which is given by the time-dependent donor-acceptor distance $r(t)$ and the dye-specific Förster-distance R'' (Supplementary Information eq. S1)." The definition of a continuous true FRET efficiency is confusing. What I believe the authors are trying to say is that, for dynamic systems, the FRET efficiency then fluctuates between states with time. However, this could theoretically also be detectable). In addition, if the orientation of the dyes (or the spectra) are changing, so is the $R0$ and then $R0$ should be time dependent. The authors need to be more explicit about the difference between ϵ (Mathematical epsilon) and ε (Greek epsilon).

We thank the referee for raising this point. Our definition of a true and continuous FRET efficiency results from the notion of classical physics that motions of atoms obey Hamilton's equations. As such, the motions, i.e., the trajectories of atoms, are inherently continuous and differentiable with respect to time. For a biomolecule that switches between two ‘states’, switching will not be instantaneous as also demonstrated more recently by the measurement of transition path times. A coarse-grained view from statistical physics would describe this switching as diffusion in a potential of mean force (PMF) with a barrier that separates two wells. The picture of instantaneous switching between discrete states that we often use in chemistry is a simplification that only holds under conditions at which the barrier is sufficiently high. In the same spirit, any temporal change of a distance in a molecule will be a continuous process at its heart. In our derivation, we started from these continuous changes and used Förster's formula to convert the distance coordinate into

the probability ε of observing a photon of this or that color at any point in time. However, we admit that equation 1 in the Supplementary Information does not account for the changes in the relative orientation of the dyes, which will also impact the probability of photon emission. As our derivation is solely based on the quantity $\varepsilon(t)$ rather than the distance $r(t)$, the derivation remains unchanged if we allow also fluctuations of the Förster distance $R_0(t)$. We now changed eq. S1 to clarify this aspect and added a brief explanation. The difference between ε and ϵ (now \tilde{E} in the revised manuscript) is that ε is a probability whereas ϵ (now \tilde{E}) is the outcome of realizations of this probability.

Figure 1a: In English, one reads from left to write. Hence, I suggest flipping the pathway of the molecule through the volume to follow accordingly. (Also in Figure 5c).

We now changed the direction of diffusion through the volume.

Figure 2a: add 'n=' to figure legends.

We added 'n =' to the figure legends.

Figure 2d: Please write 2k out as $k_{12} + k_{21}$ to be less confusing.

We now write $k_{12} + k_{21}$ in Figure 2d.

Lines 173-174: "In fact, static heterogeneity is difficult to spot otherwise and is rarely included in model-based analysis approaches." - E-tau plots are useful for detecting static and dynamic heterogeneities. Please rewrite.

We now mention the usefulness of E-tau plots at this point (p. 6).

Figure 4d-j. A discussion of more realistic parameters and mentioning the various values for typical dye pairs would help improve this figure.

The parameters used by us encompass the realistic parameters as we used an extreme range of gamma from 0.1 to 10. However, we now highlight explicitly the realistic parameter range range of 0.5 to 2 in Fig. 4f-g.

Figure 4d, 4h. What is the meaning of the grey boxes? I find it confusing that the box narrows when the brightnesses don't change with protein conformation, and retains the same width when the brightnesses do change with conformation.

We apologise for the confusion. The gray boxes only had the intention to highlight the parameters. We now made the gray boxes identical in shape and size in Figure 4d and 4h.

Figure 5a: For the transition from S0S1 to S1S1. The up arrow should be dark blue.

We thank the referee for spotting this error. We now corrected the color of the transition arrow.

Figure 5a, caption: "The first letter refers to the donor and the second letter indicates the state of the acceptor." This is redundant with respect to a previous sentence: "Each state is denoted by the electronic states of donor (first symbol) and acceptor (second symbol)"

The referee is absolutely correct and we removed this redundancy in the revised manuscript.

Figure 5d: Which FRET efficiency (i.e. which epsilon) is being plotted here. Not always easy to follow in the text.

We now state explicitly that we plot the uncorrected FRET efficiency in Figure 5d, i.e., the quantity E . We also went over the whole manuscript to clarify the type of FRET efficiency in the axis labels.

Figure 5e, inset: The abscissa axes should be labeled 'Lag Time'.

We thank the referee for spotting this mistake and corrected it in the revised version.

Figure 5e, caption: This looks very much like nsFCS, but a couple orders of magnitude slower. It would be useful to explicitly highlight the difference in timescale here.

We now highlight the difference in timescale to nsFCS. We additionally point out that every correlation function can be represented with positive and negative lag times. This is unnecessary for auto-correlation functions. Yet, for cross-correlation functions, the presentation with positive and negative lag times is helpful to spot the breakage of time-reversal symmetry, such as for acceptor bleaching. However, we also would like to note that the reverse is not necessarily true, i.e., not every breakage of time reversal symmetry leads to an asymmetric crosscorrelation branches.

The authors talk about "bleaching" throughout the paper. This should be referred to as "photobleaching".

We now replace bleaching by photobleaching throughout the manuscript.

Figure 6b: Can the triplet state be extracted from these plots and then used to fix the timescale of the triplet kinetics in panel c?

Unfortunately, this is not possible. The timescale of 8 μ s for the fast component and also the high amplitude compared to our simulations indicate that triplet and conformational dynamics mix.

Figure 6c. Please plot both green components and provide the uncertainty in the rates.

We now provide the both green components provide the fitting errors of the rates in the revised figure (now Fig. 7).

Figure 6c,d: Please include details of how the fitting was performed. The 1000 label is shifted in the abscissa axis.

We use the standard routine of minimizing least squares in all exponential fits using the Mathematica version 13.2. We now clearly state that in the Methods (p. 27). We also corrected the position of the axis label.

Lines 487-491: “We collected > 800,000 bursts to also probe the nanosecond dynamics sufficient photon pair statistics (Fig. 8a, inset).” The (Fig 8a, inset) is confusing. The nanosecond dynamics are not in the inset, only $n = 847,012$. Hence, the reference is not clear.

We thank the referee for spotting this inaccuracy in our writing and now corrected it.

Line 575: Please give the diffusion coefficient in $\mu\text{m}^2/\text{s}$.

We now provide the diffusion coefficient in the units preferred by the referee.

Lines 619-620: Please give background rates in Hz (or kHz).

We also provide the background rates in kHz.

Line 624: Here, the authors introduce crosstalk, but do not refer to it in the development of the theory. Please discuss the influence of crosstalk when correcting the correlation function for system imperfections.

We would like to point out that FRET correlation functions are always the correlation functions of the uncorrected FRET efficiency. Hence, we do not correct FRET correlation functions. To understand how cross-talk impacts FRET correlation functions, we performed additional simulations at different percentages of leakage of the donor signal into the acceptor channel in Supplementary Fig. 6. We found that cross-talk reduces the amplitude of the FRET correlation functions, as expected. In the limit of 100% crosstalk, the signals of donor and acceptor would be identical and perfectly correlated, which erases all changes in the FRET efficiency.

Line 669: “Here, n is the total photon emission rate of donor and acceptor.” This is unclear. Is this the sum of the donor and acceptor emission rates, or will there be subscripts to denote donor and acceptor? Please clarify.

We now clarify that n is the sum of the donor and acceptor photon rate (p. 22).

Lines 675-677: “To minimize the noise in the recolored FRET correlation functions, we recolored the data ten times and computed the average FRET correlation function of the ten recolored data sets.” I do

not follow, how does this minimize the noise? If this is the same data, how does recoloring it ten times help?

As explained in our reply to referee 2, the recoloring process involves the generation of a trajectory of states for each burst followed by generating new outcomes for the photons detected in the burst based on the state. Both, the generation of the trajectory and rolling a die to determine the photon color are stochastic outcomes and therefore carry intrinsic noise. We minimize this noise by performing the procedure ten times and then average the resulting FRET histograms. In the revised manuscript, we indicate the variations associated to the recoloring procedure as vertical bars in Fig. 3a and c. The procedure is now explained in more detail in the methods section of the revised manuscript.

Lines 691-692: "To access the effect of bleaching (Fig. 4a-c), we modelled a system that switches between 4 states:" I assume that switching from the photobleached states back to FRET states is not allowed. Please reword.

Indeed, the referee is correct. Photobleaching in this simulation is irreversible. We now changed the statement to clarify this point (p. 23).

Line 710: " $\gamma_{PIE} = 2$ " Please define what this is.

The definition of the value γ_{PIE} , which is the factor required to obtain a FRET population with a stoichiometry value of 0.5, was defined in eq. 26. We now refer to this equation in the revised manuscript.

Line 724: Typo: form should be from.

We thank the referee for spotting this typo and corrected in the revised version.

Line 881: Equation 27. The authors should site the work where this was introduced over 10 years ago.

We now cite the paper of Kudryavtsev *et al.* (2012) who introduced a similar criterion.

Line 987: Typo in reference: Dueller should read Mueller. (Also SI Reference 3).

We thank the referee for spotting this typo and corrected it in the revised version.

SI: Throughout the SI, please add subscripts to the $\langle \rangle$ to clarify what dimension is being averaged over.

We thank the referee for the suggestion. However, we would prefer not to add unnecessary subscripts. We clearly specified that the brackets $\langle \dots \rangle$ indicate ensemble averages. In extreme cases, we already added the subscript $\langle \dots \rangle_{\tau}$ to indicate that the average is lag time dependent. For all other cases, the meaning of the bracket is specified in the text.

SI Line 220-222: Eqn S26: Please add subscript D to sigma.

We follow the referee's advice and add the subscript D to the sigma.

SI Line 366: Typo. Should read "Fig. 1..."

We corrected this mistake.

SI Line 444: Typo: "Indepent" -> Independent

We thank the reviewer and corrected the typo.

1. Gopich, I.V. & Szabo, A. Single-molecule FRET with diffusion and conformational dynamics. *The journal of physical chemistry B* **111**, 12925-32 (2007).
2. Song, K., Makarov, D.E. & Vouga, E. Information-theoretical limit on the estimates of dissipation by molecular machines using single-molecule fluorescence resonance energy transfer experiments. *J Chem Phys* **161**(2024).
3. Agam, G. et al. Reliability and accuracy of single-molecule FRET studies for characterization of structural dynamics and distances in proteins. *Nat Methods* **20**, 523-535 (2023).
4. Pirchi, M. et al. Photon-by-Photon Hidden Markov Model Analysis for Microsecond Single-Molecule FRET Kinetics. *J Phys Chem B* **120**, 13065-13075 (2016).
5. Vollmar, L., Schimpf, J., Hermann, B. & Hugel, T. Cochaperones convey the energy of ATP hydrolysis for directional action of Hsp90. *Nat Commun* **15**, 569 (2024).
6. Igoshin, O.A., Kolomeisky, A.B. & Makarov, D.E. Uncovering dissipation from coarse observables: A case study of a random walk with unobserved internal states. *J Chem Phys* **162**(2025).